# Tumor exosome-based nanoparticles are efficient drug carriers for chemotherapy

Tuying Yong [1,2,6], Xiaoqiong Zhang[1,6], Nana Bie[1,6], Hongbo Zhang[3,6], Xuting Zhang [1], Fuying Li[1], Abdul Hakeem[1], Jun Hu[1], Lu Gan [1,2], Hélder A. Santos [4,5] & Xiangliang Yang[1,2]

Developing biomimetic nanoparticles without loss of the integrity of proteins remains a major challenge in cancer chemotherapy. Here, we develop a biocompatible tumor-cell-exocytosed exosome-biomimetic porous silicon nanoparticles (PSiNPs) as drug carrier for targeted cancer chemotherapy. Exosome-sheathed doxorubicin-loaded PSiNPs (DOX@E-PSiNPs), generated by exocytosis of the endocytosed DOX-loaded PSiNPs from tumor cells, exhibit enhanced tumor accumulation, extravasation from blood vessels and penetration into deep tumor parenchyma following intravenous administration. In addition, DOX@E-PSiNPs, regardless of their origin, possess significant cellular uptake and cytotoxicity in both bulk cancer cells and cancer stem cells (CSCs). These properties endow DOX@E-PSiNPs with great in vivo enrichment in total tumor cells and side population cells with features of CSCs, resulting in anticancer activity and CSCs reduction in subcutaneous, orthotopic and metastatic tumor models. These results provide a proof-of-concept for the use of exosome-biomimetic nanoparticles exocytosed from tumor cells as a promising drug carrier for efficient cancer chemotherapy.

[1] National Engineering Research Center for Nanomedicine, College of Life Science and Technology, Huazhong University of Science and Technology, 430074 Wuhan, China. [2] Hubei Key Laboratory of Bioinorganic Chemistry and Materia Medica, Huazhong University of Science and Technology, 430074 Wuhan, China. [3] Pharmaceutical Sciences Laboratory and Turku Center for Biotechnology, Åbo Akademi University, 20520 Turku, Finland. [4] Drug Research Program, Division of Pharmaceutical Chemistry and Technology, University of Helsinki, FI-00014 Helsinki, Finland. [5] Helsinki Institute of Life Science, University of Helsinki, FI-00014 Helsinki, Finland. [6] These authors contributed equally: Tuying Yong, Xiaoqiong Zhang, Nana Bie, Hongbo Zhang. Correspondence and requests for materials should be addressed to L.G. (email: lugan@mail.hust.edu.cn) or to H.A.S. (email: helder.santos@helsinki.fi) or to X.Y. (email: yangxl@mail.hust.edu.cn)

Nanoparticles-based drug delivery systems (NDDSs) have shown promising therapeutic efficacy in cancer due to the enhanced permeability and retention (EPR) effect[1,2]. To increase the capacity of targeting delivery of anticancer drugs to tumors, nanoparticles are usually functionalized with targeted antibodies, peptides or other biomolecules[3,4]. However, the presence of targeting ligands may sometimes have a negative influence on nanoparticle delivery owing to the enhanced immune-elimination[5]. Moreover, the targeting of these functionalized nanoparticles using targeting ligands is not possible and not precise for a wide range of cancers, because the receptors differ from versatile genetic or phenotypic heterogeneity of tumors[6,7].

Biomimetic nanoparticles that combine the unique functionalities of natural biomaterials, such as cells or cell membranes, and engineering versatility of synthetic nanoparticles have recently increased considerable attention as effective drug delivery platforms[8,9]. Nanoparticles can be coated by various cell membranes from red blood cells (RBCs)[10,11], cancer cells[12,13], platelets[14], or white blood cells (WBCs)[15], and have displayed good biocompatibility, prolonged circulation, as well as tumor-targeting capacity. Exosomes are small extracellular vesicles secreted by mammalian cells[16], and have lately been used as attractive nanocarriers owing to their stability in circulation, biocompatibility, low immunogenicity and low toxicity[17–19]. Furthermore, the exosomes display efficient cellular uptake and target-homing capabilities dependent on the proteins of their membrane[17–19]. Given that the surface protein composition of exosomes may be crucial to their function, preservation of exosome membrane integrity and stability is very important for their application in drug delivery[20]. Generally, exosomes-biomimetic nanoparticles are constructed by iterative physical extrusion or freeze/thaw cycles to fuse exosomes and nanoparticles[21,22], which might affect the protein integrity on exosome membranes, thereby compromising the biofunctions of these biomimetic nanoparticles[23,24]. Therefore, it is highly desired to develop an efficient approach to construct exosome-biomimetic nanoparticles without interfering with the membrane integrity for cancer therapy.

Luminescent porous silicon nanoparticles (PSiNPs) have been widely used as drug carriers owing to their excellent drug loading capacity, high biocompatibility and biodegradability[15,25–30]. Here, we develop a biocompatible tumor cell-exocytosed exosome-sheathed PSiNPs (E-PSiNPs) as a drug carrier for targeted cancer chemotherapy. When tumor cells are incubated with doxorubicin-loaded PSiNPs (DOX@PSiNPs), they exocytose exosome-sheathed DOX-loaded PSiNPs (DOX@E-PSiNPs) (Fig. 1a). DOX@E-PSiNPs, regardless of the exosome origin, possess strong cross-reactivity of cellular uptake and cytotoxicity against bulk cancer cells and cancer stem cells (CSCs), which are responsible for tumorgenesis, tumor progression, recurrence, metastasis and drug resistance[31,32]. Moreover, DOX@E-PSiNPs exhibit enhanced tumor accumulation, extravasation from blood vessels and deep penetration into tumor parenchyma. These features of DOX@E-PSiNPs result in their greater in vivo enrichment in total tumor cells and side population cells with characteristics of CSCs[33,34], thus generating remarkable anticancer and CSCs killing activity in subcutaneous, orthotopic and metastatic tumors (Fig. 1b). Our study provides a approach for cancer therapy by using exosome-biomimetic nanoparticles exocytosed from tumor cells as drug carriers to efficiently deliver anticancer drug.

## Results

### Autophagy-involved in the exocytosis of PSiNPs. Luminescent PSiNPs were prepared by electrochemical etching of silicon

wafers, lift-off of PSi film, ultrasonication, centrifugation and finally activation of luminescence by heating in an aqueous solution. The hydrodynamic diameter of PSiNPs was ca. 150 nm measured by dynamic light scattering (DLS, Supplementary Fig. 1a). Scanning electron microscope (SEM) image showed a meso-porous nanostructure of the PSi film with the pore diameter of ca. 11 nm (Supplementary Fig. 1b). The BET surface area, pore volume and average pore diameter of PSiNPs were 211.8 m$^2$ g$^{-1}$, 0.2 cm$^2$ g$^{-1}$, and 13.5 nm as measured by nitrogen adsorption analysis (Supplementary Fig. 1c), respectively. The intrinsic photoluminescence of PSiNPs under 488 nm excitation appeared at wavelengths between 600 and 800 nm (Supplementary Fig. 1d).

Human hepatocarcinoma Bel7402 cells were treated with PSiNPs for 6 h, followed by washing thoroughly with PBS and then incubating in fresh nanoparticle-free medium for different time intervals. Inductively coupled plasma-optical emission spectroscopy (ICP-OES) analysis showed that PSiNPs were exocytosed from Bel7402 cells in a time-dependent manner, and ca. 96% of PSiNPs were expelled out after culture in fresh medium for 18 h (Supplementary Fig. 2). Autophagy is a highly regulated process for intracellular homeostasis through clearance, degradation, or exocytosis of damaged cell components or foreign risks[35]. Thus, the exocytosis of PSiNPs may have high relevance to autophagy. To elucidate the role of autophagy in the exocytosis of PSiNPs, we first sought to determine whether PSiNPS-induced autophagy. Bel7402 cells were treated with PSiNPs for different time intervals and then the ratio of endogenous microtubule-associated protein 1 light chain 3 (LC3)-II to LC3-I was assessed, since cytosolic LC3-I is conjugated to phosphatidylethanolamine to form membrane-associated LC3-II, which is recruited to autophagosomal membranes during autophagy, and therefore the conversion of LC3-I to LC3-II is considered to be an accurate indicator of autophagic activity[36]. PSiNPs treatment resulted in an increase of the ratio of LC3-II to LC3-I in a time-dependent manner (Fig. 2a), suggesting that PSiNPs treatment induces a cumulative increase in the formation of autophagosomes. Similar result was observed in murine hepatocarcinoma H22 cells (Supplementary Fig. 3), revealing that this phenomenal was not cell dependent. Bel7402 cells were also transfected with EGFP-LC3 plasmid and then treated with PSiNPs for different time intervals. Consistently, treatment with PSiNPs led to significantly enhanced puncta formation of LC3-labeled vacuoles (Fig. 2b), confirming that PSiNPs-induced autophagosome formation. Moreover, we observed intracellular PSiNPs captured in the LC3$^+$ autophagosomes (Fig. 2b). To determine whether autophagy was involved in the exocytosis of PSiNPs, we exposed Bel7402 cells with PSiNPs in the presence or absence of autophagy inhibitor 3-methyladenine (3-MA), or autophagy inducers rapamycin and carbamazepine (CBZ). 3-MA significantly inhibited the exocytosis of PSiNPs, while both rapamycin and CBZ significantly enhanced the exocytosis of PSiNPs (Fig. 2c), indicating that autophagy mediates the exocytosis of PSiNPs. Furthermore, the exocytosis of PSiNPs from Atg7 (a crucial autophagy gene)-deficient (Atg7$^{-/-}$) mouse embryonic fibroblasts (MEFs) was significantly lower than that from wild type MEFs (Fig. 2d), confirming that PSiNPs-induced autophagy regulates their exocytosis after internalization.

### Exosomes sheathed with PSiNPs (E-PSiNPs). After Bel7402 cells were incubated with PSiNPs, we collected the exocytosed PSiNPS (E-PSiNPs) by centrifugation. Field transmission electron microscope (FTEM) energy spectrum analysis showed that silicon was detected in E-PSiNPs (Supplementary Fig. 4), endorsing that E-PSiNPs were actually the exocytosed PSiNPs. DLS analysis showed that the size of E-PSiNPs and PSiNPs was 260 ± 15 nm

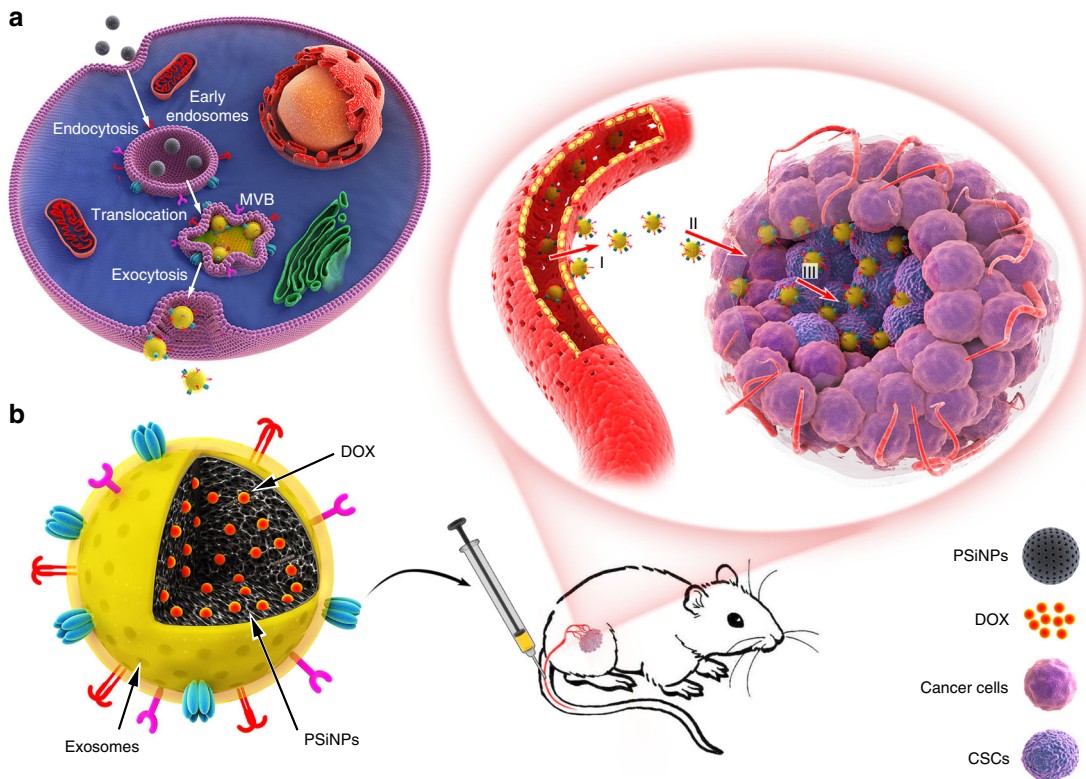

**Fig. 1** Schematic illustration of E-PSiNPs as drug carriers for targeted cancer chemotherapy. **a** Schematic illustration of the preparation of DOX@E-PSiNPs. DOX@PSiNPs are endocytosed into cancer cells after incubation, then localized in multivesicular bodies (MVBs) and autophagosomes. After MVBs or amphisomes fuse with cell membrane, DOX@PSiNPs are exocytosed into extracellular space. **b** Schematics showing how DOX@E-PSiNPs effeciently target tumor cells after intravenous injection into tumor-bearing mice. (I) DOX@E-PSiNPs effeciently accumulate in tumor tissues; (II) DOX@E-PSiNPs penetrate deeply into tumor parenchyma; and (III) DOX@E-PSiNPs are efficently internalized into bulk cancer cells and CSCs to produce strong anticancer efficacy

and $150 \pm 11$ nm, and the corresponding PDI was $0.145 \pm 0.032$ and $0.208 \pm 0.028$, respectively (Fig. 3a). The zeta-potential of E-PSiNPs and PSiNPs was $-11.0 \pm 0.4$ mV and $-10.8 \pm 0.2$ mV. TEM images revealed that PSiNPs and E-PSiNPs displayed irregular morphology, and ca. 20 nm thick membrane appeared on the surface of E-PSiNPs compared with PSiNPs (Fig. 3b). To further prove that PSiNPs were sheathed with membrane structure in E-PSiNPs, 3,3′-dioctadecyloxacarbocyanine perchlorate (DiO), a commonly used cell membrane fluorescent probe, was used to stain E-PSiNPs. Colocalization of green DiO fluorescence with intrinsic red PSiNPs fluorescence was observed in E-PSiNPs, but not in PSiNPs by confocal microscopy (Fig. 3c), confirming the presence of the membrane sheathed on PSiNPs in E-PSiNPs.

Exosomes, derived from fusion of intraluminal vesicles in MVBs with plasma membrane, serve as highly efficient export vehicles[17–19]. When internalized into Bel7402 cells, PSiNPs were found to be colocalized with FITC-CD63-labeled MVBs (Supplementary Fig. 5), suggesting that PSiNPs are associated with MVBs before exocytosis. To explore whether E-PSiNPs were sheathed with exosomes, FITC-conjugated CD63 (a common biomarker for exosomes) antibody was used to label E-PSiNPs exocytosed from Bel7402 cells. As shown by immunofluorescent staining, CD63 was detected in E-PSiNPs, but not in PSiNPs (Fig. 3d). Western blot experiments further showed that similar to the whole cell lysates and the purified exosomes obtained by differential ultracentrifugation[37,38], exosome biomarkers TSG101 and CD63 were also detected in E-PSiNPs (Fig. 3e), confirming the presence of exosomes in E-PSiNPs. In contrast to exosome biomarkers, calnexin, a protein located in endoplasmic reticulum (ER)[39], was only detected in whole cell lysates, but not in both

E-PSiNPs and the purified exosomes (Fig. 3e), revealing the high purity of the exosomes sheathed on PSiNPs in E-PSiNPs. Similar results were also observed in E-PSiNPs exocytosed from H22 cells (Supplementary Fig. 6), suggesting that E-PSiNPs can be generated from different cell lines. Moreover, dimethyl amiloride (DMA), an inhibitor of exosome release by disrupting calcium signaling[40], was found to significantly inhibit the yield of E-PSiNPs, while ionomycin, a promoter of exosome release by increasing intracellular calcium concentration[41], significantly augmented the yield of E-PSiNPs (Fig. 3f).

Overall, these results strongly confirm that the membrane that sheathed PSiNPs in E-PSiNPs is exosomes. The total protein amount of E-PSiNPs exocytosed from $10^7$ cancer cells was 60 μg, but only 1.8 μg proteins were detected in the naturally secreted exosomes from equal numbers of cancer cells by differential ultracentrifugation, which was consistent with the other group's report[21]. The fact that PSiNPs stimulated the production of exosomes by nearly 34 times shows that E-PSiNPs can be prepared with relatively high yield.

**E-PSiNPs as a drug carrier**. E-PSiNPs as a drug carrier were investigated using DOX as a model drug. DOX was loaded into PSiNPs and then incubated with Bel7402 cells. Exosome-sheathed DOX-loaded PSiNPs (DOX@E-PSiNPs) were also obtained by centrifugation in a similar fashion to E-PSiNPs. The colocalization of DOX, DiO-labeled membrane and PSiNPs by immunofluorescent staining showed the successful encapsulation of DOX into E-PSiNPs exocytosed from Bel7402 cells (Fig. 4a). DOX can also be encapsulated into E-PSiNPs exocytosed from H22 cells

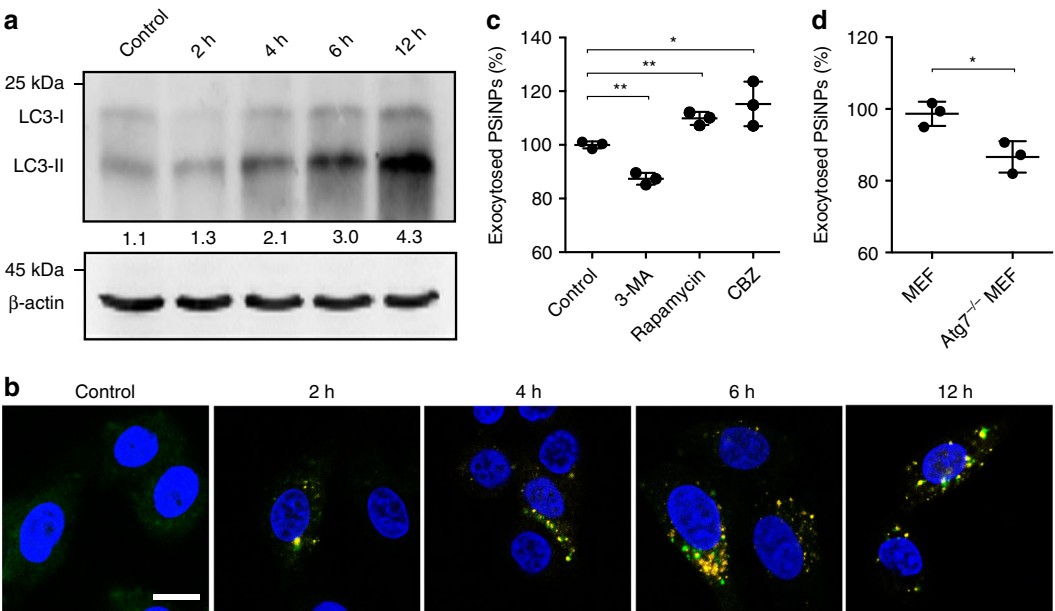

**Fig. 2** Role of autophagy in the exocytosis of PSiNPs. **a** LC3-I and LC3-II expression in Bel7402 cells treated with 200 μg mL$^{-1}$ PSiNPs for different time intervals by western blot. The number underneath each group in the immunoblotting indicates the relative ratio of LC3-II to LC3-I of the corresponding group. **b** Confocal fluorescence microscopic images of EGFP-LC3-transfected Bel7402 cells after treatment with 200 μg mL$^{-1}$ PSiNPs for different time intervals. Scale bar: 20 μm. **c** Relative amount of the exocytosed PSiNPs in Bel7402 cells after treatment with 200 μg mL$^{-1}$ PSiNPs for 6 h, followed by washing with PBS and then incubating in fresh medium with or without 5 mM of 3-MA, 200 nM of rapamycin or 30 μM of CBZ for another 16 h by ICP-OES. **d** Relative amount of the exocytosed PSiNPs in wild type and Atg7$^{-/-}$ MEF cells after treatment with 200 μg mL$^{-1}$ PSiNPs for 6 h, followed by washing with PBS and then incubating in fresh medium for 16 h by ICP-OES. Data were represented as mean ± SD ($n = 3$). *$P < 0.05$, **$P < 0.01$ (one-way ANOVA with Fisher's LSD test for **c** and unpaired two-tailed Student's $t$ test for **d**). Source data are provided as a Source Data file

using the same processing method (Supplementary Fig. 7). The drug loading degree of DOX@E-PSiNPs was 300 ng DOX μg$^{-1}$ protein (exosomes were quantified according to the protein content) and the drug loading efficiency was 0.8% determined by high performance liquid chromatography (HPLC). DOX loading did not significantly change the size of E-PSiNPs (Fig. 4b). Moreover, the size of DOX@E-PSiNPs remained almost constant even after incubating in PBS with or without 10% fetal bovine serum (FBS) for 6 days (Fig. 4c). Furthermore, storage at −80 °C for 1 month or lyophilization followed by resuspension in PBS 1 week later did not affect the size (Supplementary Fig. 8a, d) and zeta-potential (Supplementary Fig. 8b, e) of DOX@E-PSiNPs. Furthermore, relatively little degradation (Fig. 4d) and no significant morphology change (Supplementary Fig. 9) of DOX@E-PSiNPs were detected after 72 h incubation in PBS. These results demonstrate that DOX@E-PSiNPs are relatively stable. DOX@E-PSiNPs showed a sustained drug release profile as compared to DOX@PSiNPs (Fig. 4e), which can avoid the side effects caused by DOX burst release during blood circulation.

**Efficient cellular uptake and cytotoxicity**. To explore the biological function of DOX@E-PSiNPs, the interaction of DOX@E-PSiNPs with CSCs with high drug resistance was first investigated. The H22 CSCs tumor spheroids were selected by the previously reported soft three-dimensional (3D) fibrin gel method[42,43]. Intracellular DOX fluorescence increased in a dose-dependent manner in H22 CSCs treated with free DOX, DOX@PSiNPs or DOX@E-PSiNPs exocytosed from H22 cells (Fig. 5a). However, DOX@E-PSiNPs displayed the highest intracellular accumulation, which was ca. 2.1 and 1.7 times more than free DOX and DOX@PSiNPs, respectively (Fig. 5a). DOX@E-PSiNPs after storage at −80 °C for 1 month or lyophilization followed by resuspension in PBS 1 week later still

exhibited similarly strong cellular uptake by H22 CSCs (Supplementary Fig. 8c, f). Furthermore, the intracellular DOX retention in H22 CSCs was determined after treatment with free DOX, DOX@PSiNPs or DOX@E-PSiNPs exocytosed from H22 cells for 2 h, followed by washing with PBS and then incubating in fresh medium for different time intervals. Treatment with DOX@E-PSiNPs resulted in the enhanced DOX retention in H22 CSCs compared with free DOX or DOX@PSiNPs (Supplementary Fig. 10a). The enhanced DOX retention in DOX@E-PSiNPs-treated H22 CSCs might be due to the decreased expression of multidrug-resistant protein P-glycoprotein (P-gp) (Supplementary Fig. 10b), a plasma membrane transporter whose expression was associated with cell membrane microenvironment[44]. DOX@E-PSiNPs-induced decrease in P-gp expression might be due to the strong interaction with cell membrane (Supplementary Fig. 11a, b), reducing the cell membrane fluidity (Supplementary Fig. 11c). Correspondingly, fewer H22 tumor spheroids were formed when H22 cells were pretreated with DOX@E-PSiNPs exocytosed from H22 cells for 4 h and then seeded in soft 3D fibrin gels (90 Pa, 400 cells per well) for 5 days as compared to those pretreated with free DOX or DOX@PSiNPs (Fig. 5b). Moreover, colony sizes were reduced significantly in DOX@E-PSiNPs-pretreated group (Fig. 5c). On the other hand, when H22 CSCs selected by soft 3D fibrin gels were treated with free DOX, DOX@PSiNPs or DOX@E-PSiNPs exocytosed from H22 cells for 24 h, DOX@E-PSiNPs also exhibited the strongest inhibition in colony number and size of tumor spheroids (Supplementary Fig. 12a, b). These results strongly suggest that DOX@E-PSiNPs display strong cellular uptake and intracellular retention with an excellent cytotoxicity against CSCs.

To evaluate whether DOX@E-PSiNPs possess cross-reactive cellular uptake and cytotoxicity, murine melanoma B16-F10 CSCs were treated with free DOX, DOX@PSiNPs or DOX@E-PSiNPs exocytosed from H22 cells. Consistently, DOX@E-

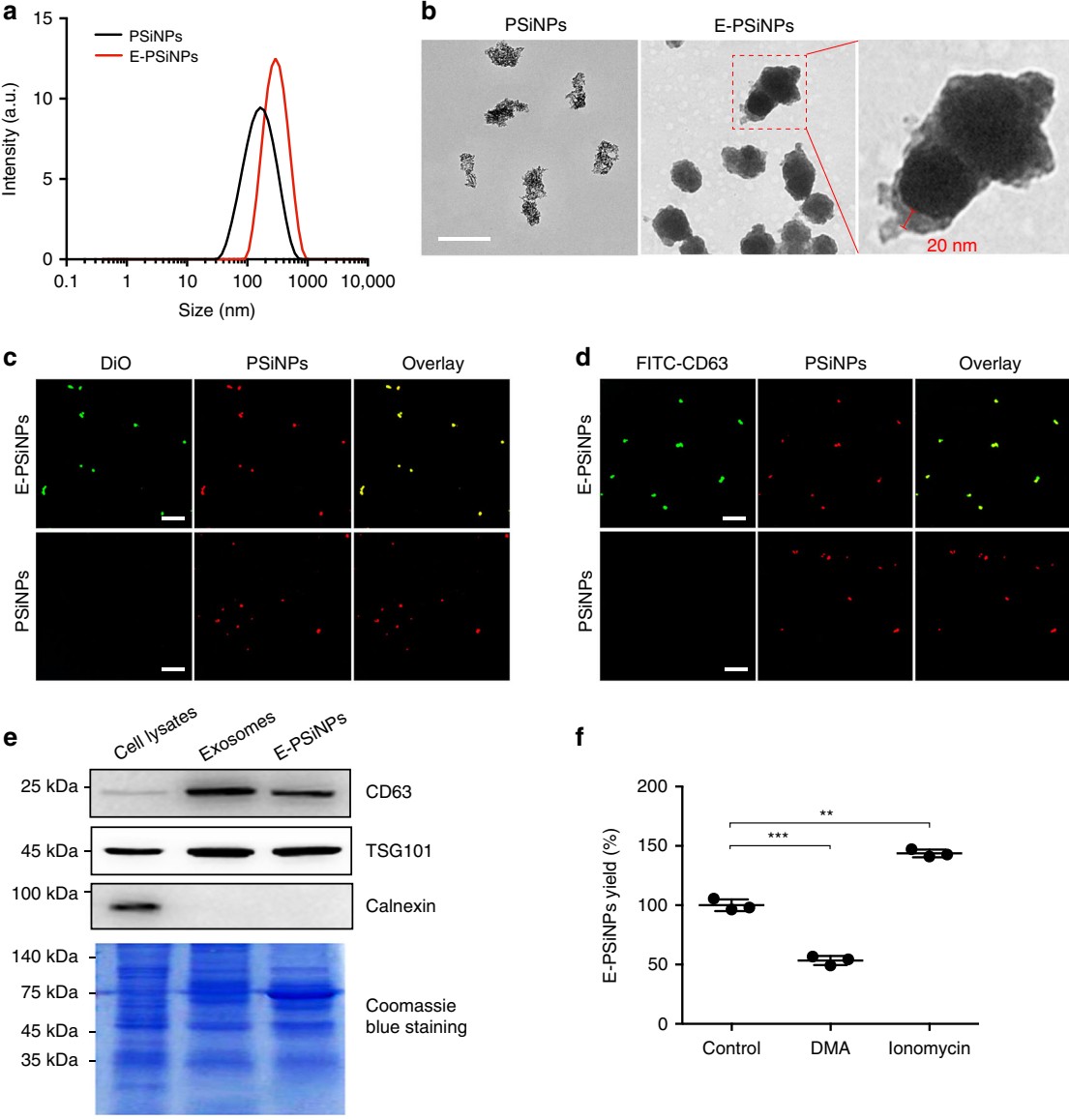

**Fig. 3** Evaluation of exosomes sheathed on PSiNPs in E-PSiNPs. **a** Hydrodynamic diameter of PSiNPs and E-PSiNPs by DLS analysis. **b** TEM images of PSiNPs and E-PSiNPs. Scale bar: 200 nm. **c** Colocalization of DiO (green) and PSiNPs (red) in E-PSiNPs by confocal microscopy. Scale bar: 20 μm. **d** Colocalization of CD63 (green) and PSiNPs (red) in E-PSiNPs by confocal microscopy. Scale bar: 20 μm. **e** Immunoblotting analysis of exosome markers (TSG101 and CD63) and ER marker (calnexin) expressed in E-PSiNPs exocytosed from Bel7402 cells. **f** Yield of E-PSiNPs when Bel7402 cells were pretreated with 200 μg mL$^{-1}$ PSiNPs for 6 h and then incubated in fresh medium containing 15 nM DMA or 10 μM ionomycin for 16 h by ICP-OES. Data were represented as mean ± SD ($n = 3$). **$P < 0.01$, ***$P < 0.001$ (one-way ANOVA with Fisher's LSD test). Source data are provided as a Source Data file

PSiNPs showed the highest internalization into B16-F10 CSCs (Fig. 5d) and had the corresponding strongest cytotoxicity against B16-F10 CSCs compared with free DOX or DOX@PSiNPs (Fig. 5e, f and Supplementary Fig. 13a, b). Similarly, DOX@E-PSiNPs exocytosed from B16-F10 cells exhibited the strongest cellular uptake and cytotoxicity against H22 CSCs (Supplementary Fig. 14). These results suggest that DOX@E-PSiNPs have a strong cross-reactive cellular uptake and cytotoxicity against CSCs, irrespective of their origin. Furthermore, DOX@E-PSiNPs also demonstrated the highest intracellular internalization and cross-reactive cytotoxicity against bulk cancer cells, such as H22, Bel7402 and B16-F10 cells compared with free DOX or DOX@PSiNPs (Supplementary Fig. 15, 16). Intracellular trafficking analysis of DOX@E-PSiNPs revealed that exosomes and DOX were internalized into cancer cells together and then colocalized with lysosomes, followed by DOX translocation to nuclei over time (Supplementary Fig. 17). Considering that more DOX was released from DOX@E-PSiNPs under lysosomal acidic pH (Supplementary Fig. 18), DOX@E-PSiNPs released DOX in lysosomes to enter nuclei to exert the cytotoxicity. CD54 (ICAM1), a member of the immunoglobulin supergene family, was found to be involved in the cross-reactive cellular uptake of DOX@E-PSiNPs by cancer cells, as evidenced by the fact that DOX@E-PSiNPs exocytosed from B16-F10 and H22 cells expressed CD54 (Supplementary Fig. 19a), and pretreatment with CD54 antibody decreased the cellular uptake of DOX@E-PSiNPs exocytosed from H22 cells by H22 and B16-F10 cells (Supplementary Fig. 19b). Despite the strong cellular uptake by tumor cells, DOX@E-PSiNPs exocytosed from H22 cells exhibited less internalization into human umbilical vein endothelial cells

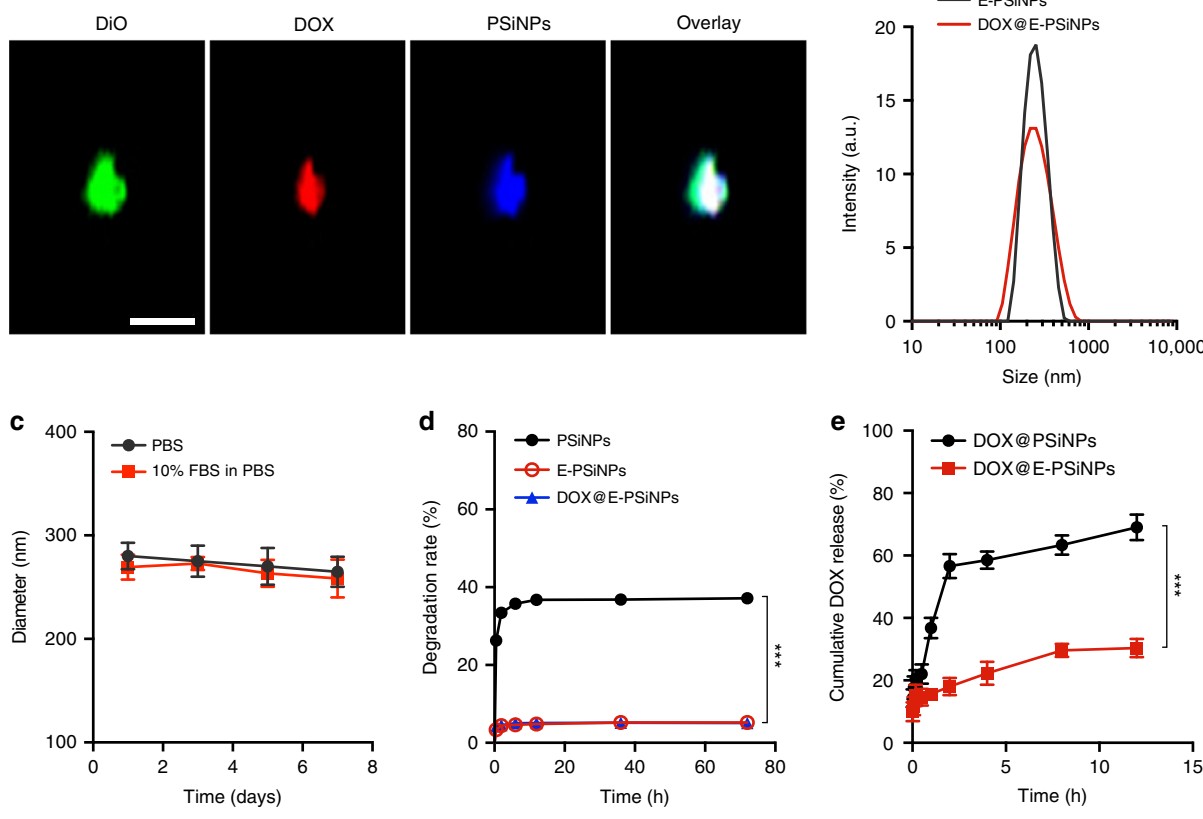

**Fig. 4** Characterization of DOX@E-PSiNPs. **a** Colocalization of DiO, DOX, and PSiNPs in DOX@E-PSiNPs exocytosed from Bel7402 cells by confocal microscopy. Scale bar: 1 μm. **b** Hydrodynamic diameter of E-PSiNPs and DOX@E-PSiNPs by DLS. **c** Hydrodynamic diameter of E-PSiNPs incubating in PBS with or without 10% FBS for different time intervals. **d** Degradation behavior of PSiNPs, E-PSiNPs and DOX@E-PSiNPs in PBS at 37 °C. **e** In vitro DOX release profiles of DOX@PSiNPs and DOX@E-PSiNPs in PBS at pH 7.4 by dialysis bag. Data were presented as mean ± SD ($n = 3$). ***$P < 0.001$ (one-way ANOVA with Bonferroni's multiple comparisons test for **d** and unpaired two-tailed Student's $t$ test for **e**). Source data are provided as a Source Data file

(HUVECs). In addition, less DOX@E-PSiNPs exocytosed from HUVECs cells were internalized into H22 cells compared with DOX@E-PSiNPs exocytosed from H22 cells (Supplementary Fig. 20), suggesting the tumor cell targeting capacity of tumor exosome-coated PSiNPs.

**Enhanced tumor accumulation and penetration.** Besides efficient cellular uptake and accompanied strong cytotoxicity against bulk cancer cells and CSCs, an ideal anticancer drug delivery system following systemic administration should be characterized by enhanced tumor accumulation and penetration to reach bulk cancer cells and CSCs. Therefore, the in vivo biodistribution of DOX@E-PSiNPs was investigated. Mice bearing H22 hepatocarcinoma tumors were intravenously injected with free DOX, DOX@PSiNPs or DOX@E-PSiNPs at 0.5 mg kg$^{-1}$ DOX dosage, or high dosage of free DOX at 4 mg kg$^{-1}$. At 24 h after injection, the tumors and major normal organs (heart, liver, spleen, lung and kidney) were collected for DOX content measurement. Although DOX@E-PSiNPs were accumulated in liver at relatively high level (Fig. 6a), especially Kupffer cells in liver (Supplementary Fig. 21), less DOX was accumulated in normal organs (heart, liver, lung and kidney) of DOX@E-PSiNPs-treated mice than that of high dosage of DOX-treated mice (Fig. 6a). However, DOX@E-PSiNPs exhibited strong tumor tropism and accumulation, ca. 2.5 and 2.3 times relative to free DOX and DOX@PSiNPs, respectively, comparable to high dosage of free DOX (Fig. 6a). Pretreatment with CD54 antibody decreased the tumor accumulation of DOX@E-PSiNPs, suggesting that similar to the cross-reactive cellular uptake by cancer cells, CD54 was also involved in the

enhanced tumor accumulation of DOX@E-PSiNPs (Supplementary Fig. 22). The strong tumor-targeting ability of DOX@E-PSiNPs was further confirmed in B16-F10 lung metastatic model (Supplementary Fig. 23).

Furthermore, we addressed the tumor penetration capacity of DOX@E-PSiNPs. First, tumor spheroids as in vivo-mimetic tumors were treated with free DOX, DOX@PSiNPs or DOX@E-PSiNPs for 24 h, and then the tumor spheroids were optically sectioned using confocal microscopy. The projection images of DOX fluorescence in tumor spheroids was reconstructed by using Amira software. DOX fluorescence intensity in both X- and Y-axis shadows was distinctly stronger in DOX@E-PSiNPs-treated group than that in free DOX- or DOX@PSiNPs-treated group at the same depth (Supplementary Fig. 24), suggesting the deep tumor penetration ability of DOX@E-PSiNPs. Furthermore, the deep tumor penetration capability of DOX@E-PSiNPs was investigated in H22 tumor-bearing mice by intravenous injection of free DOX, DOX@PSiNPs or DOX@E-PSiNPs. Confocal fluorescence microscopic images clearly showed that DOX@E-PSiNPs were distributed widely in whole tumor section at 24 h after injection (Fig. 6b). In contrast, DOX@PSiNPs and free DOX accumulated more around the blood vessels as indicated by stronger co-localization with FITC-CD31-labeled endothelial cells (Fig. 6b). The distance-dependent DOX fluorescence intensity also confirmed that only fluorescence signal of DOX delivered with DOX@E-PSiNPs was detectable at ca. 400 μm far away from the blood vessels, while free DOX or DOX delivered with DOX@PSiNPs was found at <120 μm away from the blood vessels (Fig. 6c). Overall, these results show that DOX@E-PSiNPs are

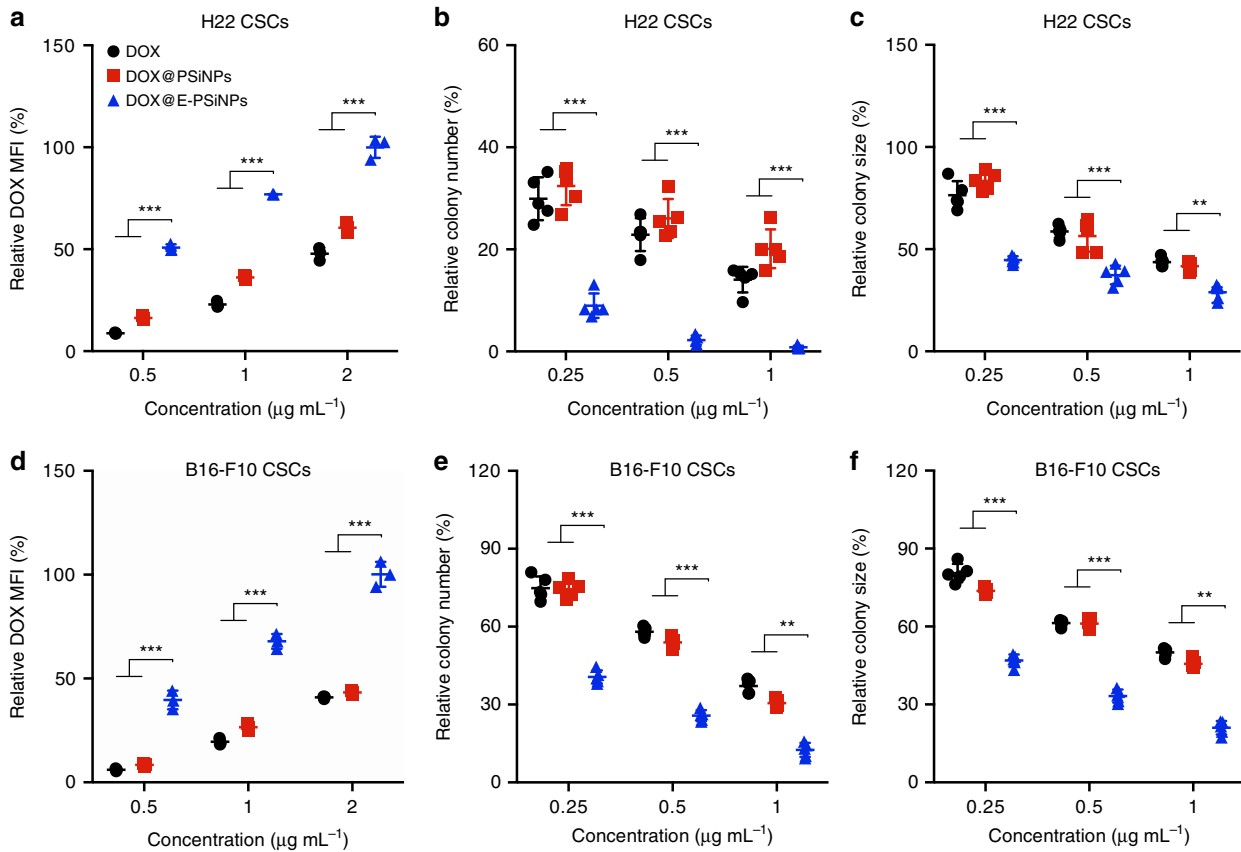

**Fig. 5** Cellular uptake and cytotoxicity of DOX@E-PSiNPs against CSCs. **a**, **d** Relative DOX mean fluorescence intensity (MFI) when H22 CSCs (**a**) and B16-F10 CSCs (**d**) selected in soft 3D fibrin gels were treated with free DOX, DOX@PSiNPs or DOX@E-PSiNPs exocytosed from H22 cells at different DOX concentrations for 2 h by flow cytometry. Data were represented as mean ± SD ($n = 3$). **b**, **e** Relative colony number of tumor spheroids when H22 (**b**) and B16-F10 cells (**e**) were pretreated with free DOX, DOX@PSiNPs or DOX@E-PSiNPs exocytosed from H22 cells at different DOX concentrations for 4 h and then seeded in soft 3D fibrin gels for 5 days. **c**, **f** Relative colony size of tumor spheroids when H22 (**c**) and B16-F10 cells (**f**) were pretreated with free DOX, DOX@PSiNPs or DOX@E-PSiNPs exocytosed from H22 cells at different DOX concentrations for 4 h and then seeded in soft 3D fibrin gels for 5 days. Data were represented as mean ± SD ($n = 5$). **P < 0.01, ***P < 0.001 (two-way ANOVA with Bonferroni's multiple comparisons test). Source data are provided as a Source Data file

easy to extravasate from the blood vessels and penetrate into deep tumor parenchyma. The strong intercellular delivery capacity of DOX@E-PSiNPs might be responsible for their enhanced tumor penetration[26] (Supplementary Fig. 25), which is regulated by CD54 expressed on exosomes of DOX@E-PSiNPs (Supplementary Fig. 26).

**Enhanced in vivo enrichment in side population cells**. Given that DOX@E-PSiNPs demonstrate enhanced tumor accumulation and penetration, as well as efficient cellular uptake by bulk cancer cells and CSCs, their in vivo enrichment in total tumor cells and CSCs might be improved. Therefore, we determined the in vivo DOX accumulation in total tumor cells at 24 h after GFP-expressing H22 tumor-bearing mice were intravenously injected with free DOX, DOX@PSiNPs or DOX@E-PSiNPs at DOX dosage of 0.5 mg kg⁻¹, or high dosage of free DOX at 4 mg kg⁻¹. The tumor tissues were digested into single cells and the intracellular DOX fluorescence in total GFP-positive tumor cells was measured by flow cytometry (Fig. 6d). DOX content in the total GFP-positive tumor cells of mice administrated with DOX@E-PSiNPs was about 3.2 times of both free DOX- and DOX@P-SiNPs-treated groups, respectively, even significantly higher than that of free DOX-treated group at high dosage. Subsequently, we isolated the side population cells from GFP-positive H22 tumors by flow cytometry and determined the intracellular DOX

fluorescence (Fig. 6e). Similarly, higher DOX fluorescence intensity was detected in side population cells of DOX@E-PSiNPs-treated mice compared with that of free DOX-, DOX@PSiNPs- or high dosage of free DOX-treated group. Collectively, these results reveal that DOX@E-PSiNPs exhibit augmented in vivo enrichment in the total tumor cells and side population cells after intravenous injection, which further increases their in vivo anticancer and CSCs killing activity.

**Excellent anticancer and CSCs killing activity**. The in vivo anticancer activity of DOX@E-PSiNPs was determined in BALB/c mice bearing subcutaneous H22 tumors. Free DOX, DOX@P-SiNPs, DOX@E-PSiNPs exocytosed from H22 cells at DOX dosage of 0.5 mg kg⁻¹, or free DOX at high dosage of 4 mg kg⁻¹ were intravenously administrated into H22 tumor-bearing mice once every 3 days for 17 days. The tumors grew very fast, and free DOX and DOX@PSiNPs at 0.5 mg kg⁻¹ dosage did not significantly inhibit the tumor growth compared with PBS and E-PSiNPs (Fig. 7a, b and Supplementary Fig. 27). In contrast, DOX@E-PSiNPs at DOX dosage of 0.5 mg kg⁻¹ showed a significant anticancer activity, with 91 and 87% reduction in tumor volume and tumor weight compared to the PBS group, respectively, and even stronger than free DOX at high dosage (Fig. 7a, b and Supplementary Fig. 27). The excellent anticancer activity of DOX@E-PSiNPs was further confirmed by increased

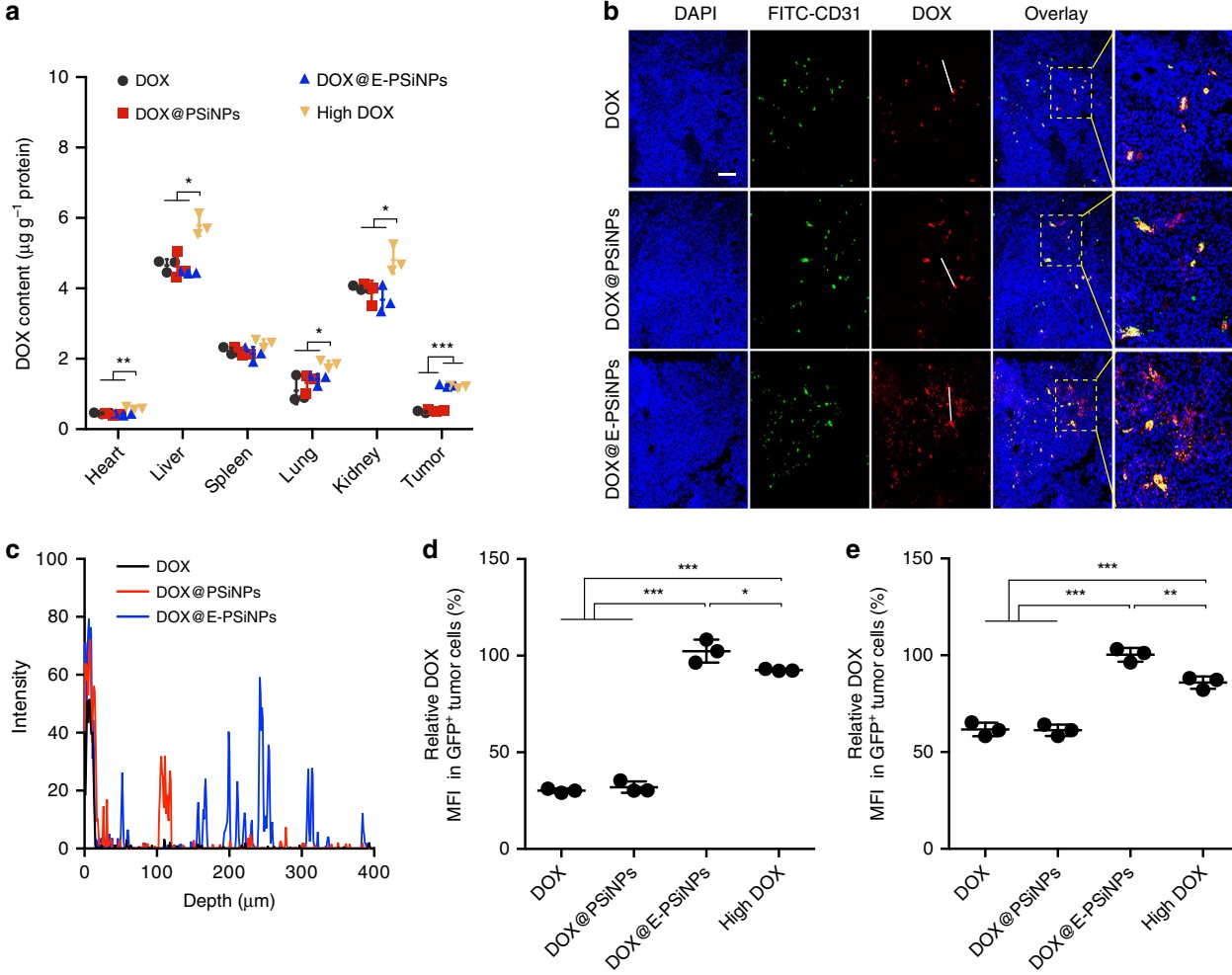

**Fig. 6** Accumulation and penetration of DOX@E-PSiNPs into tumor parenchyma. **a** DOX content in tumor tissues and major organs of H22 tumor-bearing mice at 24 h after intravenous injection of DOX, DOX@PSiNPs or DOX@E-PSiNPs at DOX dosage of 0.5 mg kg$^{-1}$, or high dosage of DOX at 4 mg kg$^{-1}$. **b** Colocalization of DOX and CD31-labeled tumor vessels in tumor sections of H22 tumor-bearing mice at 24 h after intravenous injection of DOX, DOX@PSiNPs or DOX@E-PSiNPs at DOX dosage of 0.5 mg kg$^{-1}$. Scale bar: 200 μm. White lines represent the distance between DOX in blood vessels and DOX in tumor parenchyma. **c** DOX distribution profile from the blood vessels to tumor tissues on the specified white lines as indicated in **b**. **d**, **e** Relative DOX fluorescence intensity in GFP-positive tumor cells (**d**) and side population cells (**e**) of tumor tissues at 24 h after GFP-expressing H22 tumor-bearing mice were intravenously injected with DOX, DOX@PSiNPs or DOX@ E-PSiNPs at DOX dosage of 0.5 mg kg$^{-1}$, or high dosage of DOX at 4 mg kg$^{-1}$. Data were represented as mean ± SD (n = 3). *P < 0.05, **P < 0.01, ***P < 0.001 (two-way ANOVA with Bonferroni's multiple comparisons test for **a** and one-way ANOVA with Bonferroni's multiple comparisons test for **d**, **e**). Source data are provided as a Source Data file

TUNEL-positive apoptotic tumor cells in excised tumor tissues (Supplementary Fig. 28). Moreover, prolonged survival time was observed in H22 tumor-bearing mice treated with DOX@E-PSiNPs (122 days), compared with PBS- (82 days), E-PSiNPs- (85 days), free DOX- (84 days) or DOX@PSiNPs-treated group (87 days) at 0.5 mg/kg DOX dosage, or free DOX at 4 mg/kg dosage (109 days) (Fig. 7c). Importantly, DOX@E-PSiNPs did not show systemic toxicity to H22 tumor-bearing mice, as evidenced by body weight (Supplementary Fig. 29), hematoxylin-eosin (H&E) staining of major organs (Supplementary Fig. 30) and serological analysis (Supplementary Fig. 31). However, free DOX at high dosage of 4 mg kg$^{-1}$ induced cardiotoxicity (Supplementary Fig. 30, 31).

To further assess whether DOX@E-PSiNPs could efficiently kill CSCs, the tumor tissues after treatment were digested into single cells, and the number of CD133-positive cells (a CSC marker of liver cancer[45]) was measured. As a result, CD133-positive cells were significantly inhibited in DOX@E-PSiNPs-treated group, compared with free DOX-, DOX@PSiNPs-, or high dosage of free

DOX-treated group (Fig. 7d). Consistently, the number of side population cells in tumor tissues of DOX@E-PSiNPs-treated GFP-expressing H22 tumor-bearing mice was the fewest compared with other groups (Fig. 7e). Furthermore, 800 single tumor cells were seeded in soft 3D fibrin gels (90 Pa) for 5 days, which was developed to select CSCs[42,43]. The fewest colony number and smallest colony size were formed in DOX@E-PSiNPs-treated group (Fig. 7f, g). The excellent CSCs killing activity of DOX@E-PSiNPs was further confirmed by subcutaneously transplanting the same amounts of tumor cells from tumor tissues after treatment into BALB/c mice (Fig. 7h). 100% of mice (6/6 mice) generated tumors at 6 days after secondary transplantation of tumor cells of PBS-, E-PSiNPs-, free DOX- or DOX@PSiNPs-treated group. However, 83% (5/6 mice) and 33% (2/6 mice) of mice generated tumors at 40 days after secondary transplantation of tumor cells of high dosage of DOX- and DOX@E-PSiNPs-treated groups, respectively. Taken together, these results strongly reveal that DOX@E-PSiNPs have excellent anticancer and CSCs killing activity. To further improve the

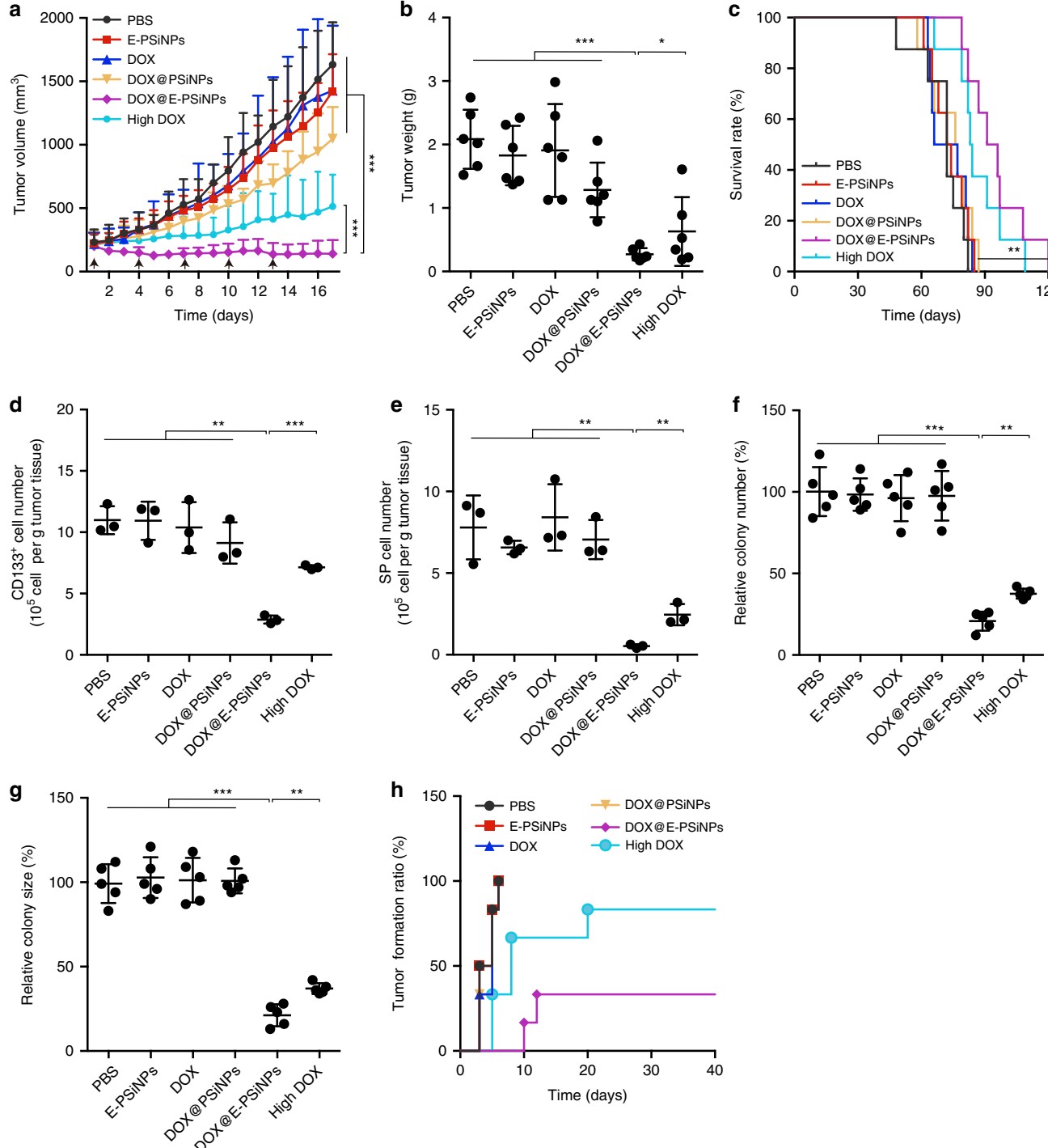

**Fig. 7** Anticancer activity of DOX@E-PSiNPs in H22 tumor-bearing mice. **a** Tumor growth curves of H22 tumor-bearing mice after intravenous injection of PBS, E-PSiNPs, free DOX, DOX@PSiNPs, DOX@E-PSiNPs exocytosed from H22 cells at DOX dosage of 0.5 mg kg$^{-1}$, or free DOX at high dosage of 4 mg kg$^{-1}$. The arrows indicate the drug injection time. Data were represented as mean ± SD ($n = 14$). **b** Weight of tumor tissues at the end of tumor growth inhibition experiments. Data were represented as mean ± SD ($n = 6$). **c** Kaplan–Meier survival plot of H22 tumor-bearing mice after intravenous administration of different formulations ($n = 8$). **d** Number of CD133-postive cells in tumor tissues at the end of tumor growth inhibition experiments. **e** Number of side population cells in GFP-positive tumor cells of GFP-expressing H22 tumor-bearing mice at the end of tumor growth inhibition experiments as above. Data were represented as mean ± SD ($n = 3$). **f, g** Relative colony number (**f**) and size (**g**) of tumor spheroids when tumor cells digested from tumor tissues of H22 tumor-bearing mice at the end of tumor growth inhibition experiments were seeded in soft 3D fibrin gels for 5 days. Data were represented as mean ± SD ($n = 5$). **h** Tumor formation ratio in BALB/c mice after subcutaneous injection of tumor cells (10$^6$ cells per mouse) from tumor tissues of H22 tumor-bearing mice after treatment as above. *$P < 0.05$, **$P < 0.01$, ***$P < 0.001$ (one-way ANOVA with Bonferroni's multiple comparisons test for **a**, **b**, and **d–g** and log-rank test for **c**). Source data are provided as a Source Data file

anticancer and CSCs killing efficacy, more DOX@E-PSiNPs at DOX dosage of 0.8 mg kg$^{-1}$, or the combination of DOX@E-PSiNPs at DOX dosage of 0.5 mg kg$^{-1}$ and all-trans-retinoic acid (ATRA), a powerful differentiating agent of CSCs, were intravenously injected into H22 tumor-bearing mice. As expected, increasing the used dosage of DOX@E-PSiNPs, or combination of DOX@E-PSiNPs and ATRA resulted in a significant tumor inhibition, with 3 or 2 tumor ablation in 6 mice, respectively (Supplementary Fig. 32a–g). Correspondingly, fewer side population cells in tumor tissues (Supplementary Fig. 32h), fewer colony number (Supplementary Fig. 32i) and smaller colony size (Supplementary Fig. 32j) after seeding the tumor cells in 3D fibrin gels were observed in these groups compared with only DOX@E-PSiNPs treatment group at DOX dosage of 0.5 mg kg$^{-1}$, suggesting that CSCs might be responsible for the drug resistance. Meanwhile, combination treatment of DOX@E-PSiNPs and ATRA, or increasing the used dosage of DOX@E-PSiNPs was found to be safe, as evidenced by routine blood test (Supplementary Fig. 33a–d), serological analysis (Supplementary Fig. 33e–j) and body weight (Supplementary Fig. 33k).

To further investigate the cross-reactive anticancer and CSCs killing efficacy of DOX@E-PSiNPs, mice bearing orthotopic 4T1 breast tumors were intravenously administrated with free DOX, DOX@PSiNPs, DOX@E-PSiNPs exocytosed from H22 cells at DOX dosage of 0.5 mg kg$^{-1}$ or free DOX at high dosage of 4 mg kg$^{-1}$ once every 3 days for 15 days. DOX@E-PSiNPs at DOX dosage of 0.5 mg kg$^{-1}$ exhibited a significant anticancer activity, with 68% and 65% reduction in tumor volume and tumor weight compared to the PBS group, respectively (Fig. 8a, b). Mice treated with DOX@E-PSiNPs had 11, 10, and 4 days longer survival time as compared to free DOX and DOX@PSiNPs at DOX dosage of 0.5 mg kg$^{-1}$, and free DOX at 4 mg kg$^{-1}$ dosage (Fig. 8c). Furthermore, tumor cells digested from breast tumors after treatment were seeded in soft 3D fibrin gels (90 Pa). The fewest colony number and smallest colony size of the formed tumor spheroids were detected in DOX@E-PSiNPs-treated group (Fig. 8d, e). These results demonstrate the excellent cross-reactive anticancer and CSCs killing efficacy of DOX@E-PSiNPs. DOX@E-PSiNPs did not cause toxicity to 4T1 tumor-bearing mice, as evidenced by routine blood test (Supplementary Fig. 34), serological analysis (Supplementary Fig. 35) and H&E staining of major organs (Supplementary Fig. 36), although free DOX at 4 mg kg$^{-1}$ dosage caused bone marrow and heart toxicity.

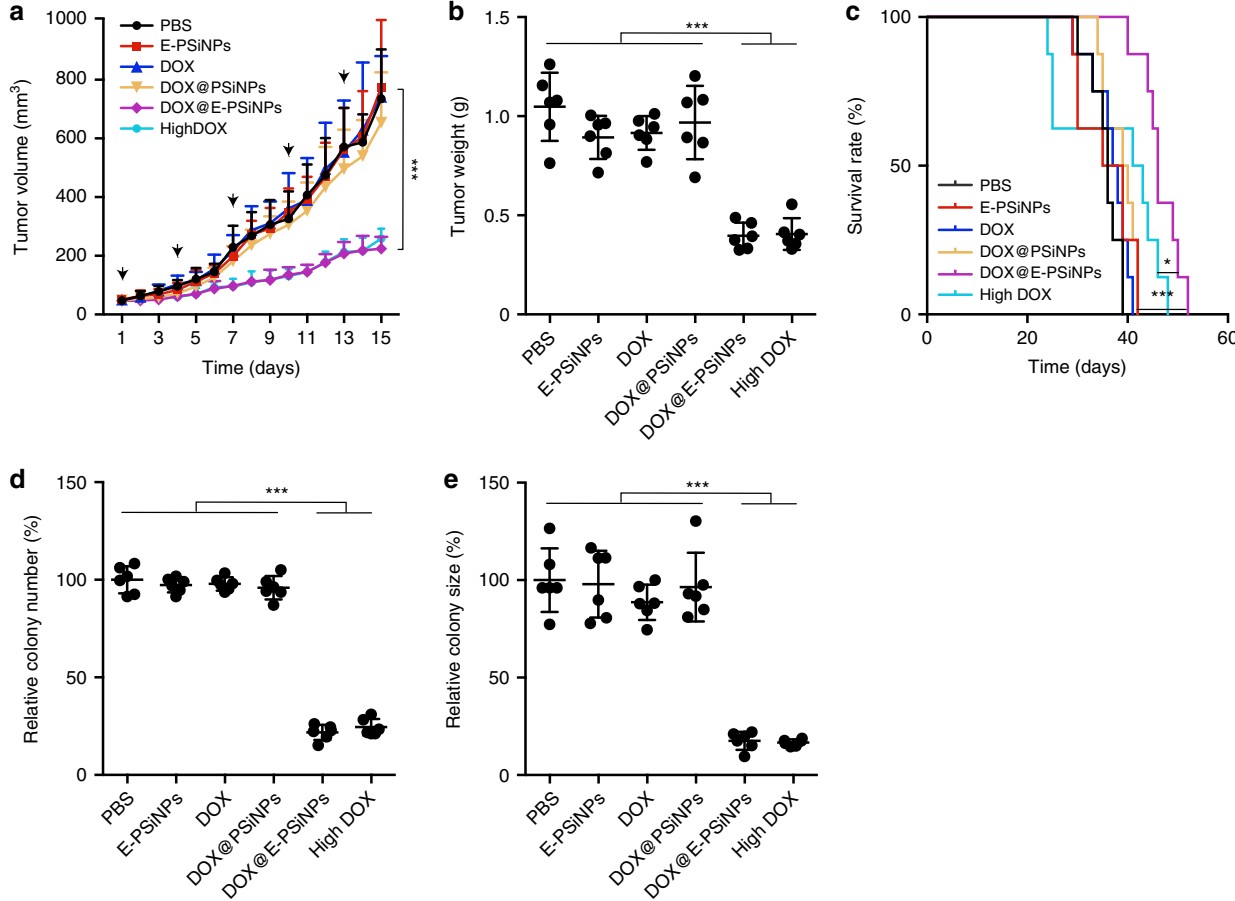

**Fig. 8** Anticancer activity of DOX@E-PSiNPs in orthotopic 4T1 tumor-bearing mice. **a** Tumor growth curves of orthotopic 4T1 tumor-bearing mice after intravenous injection of PBS, E-PSiNPs, free DOX, DOX@PSiNPs, DOX@E-PSiNPs exocytosed from H22 cells at DOX dosage of 0.5 mg kg$^{-1}$, or free DOX at high dosage of 4 mg kg$^{-1}$. The arrows indicate the drug injection time. Data were represented as mean ± SD ($n = 14$). **b** Weight of tumor tissues at the end of tumor growth inhibition experiments. Data were represented as mean ± SD ($n = 6$). **c** Kaplan–Meier survival plot of 4T1 tumor-bearing mice after intravenous administration of different formulations ($n = 8$). **d, e** Relative colony number (**d**) and size (**e**) of tumor spheroids when tumor cells digested from tumor tissues of 4T1 tumor-bearing mice at the end of tumor growth inhibition experiments were seeded in soft 3D fibrin gels for 5 days. Data were represented as mean ± SD ($n = 6$). *$P < 0.05$, ***$P < 0.001$ (one-way ANOVA with Bonferroni's multiple comparisons test for **a, b, d, e** and log-rank test for **c**). Source data are provided as a Source Data file

Furthermore, the mice model bearing B16-F10 melanoma with high lung metastasis was developed to evaluate the cross-reactive anticancer and CSCs killing activity of DOX@E-PSiNPs. At 48 h after injection of $5 \times 10^5$ B16-F10 cells into C57BL/6 mice, the mice were intravenously administrated with free DOX, DOX@P-SiNPs, DOX@E-PSiNPs exocytosed from H22 cells at DOX dosage of 0.5 mg kg$^{-1}$ or free DOX at high dosage of 4 mg kg$^{-1}$ once every 3 days for 13 days. Significantly fewer metastatic nodules were detected in the DOX@E-PSiNPs-treated group (Fig. 9a and Supplementary Fig. 37). The less lung metastasis in DOX@E-PSiNPs-treated mice was further confirmed by H&E staining on lungs (Fig. 9b). Mice treated with DOX@E-PSiNPs had 18, 17, and 4 days longer survival time as compared to free DOX and DOX@PSiNPs at DOX dosage of 0.5 mg kg$^{-1}$, and free DOX at 4 mg kg$^{-1}$ dosage (Fig. 9c). Furthermore, the fewest colony number and smallest colony size of the formed tumor spheroids were detected in DOX@E-PSiNPs-treated group after seeding the tumor cells digested from lungs in the 3D fibrin gels

(Fig. 9d, e). These results strongly demonstrate the excellent anticancer and CSCs killing efficacy of DOX@E-PSiNPs, regardless of tumor models used and the origin of exosomes used in DOX@E-PSiNPs. The cross-reactive anticancer treatment of DOX@E-PSiNPs did not induce immunological reaction, as evidenced by the fact that treatment with DOX@E-PSiNPs exocytosed from H22 cells did not affect the content of IgM, TNF-α, IL-1β, and IL-6 in serum of C57BL/6 mice (Supplementary Fig. 38).

## Discussion

CSCs, a small population of cancer cells with self-renewal and high tumorigenesis, play an important role in tumor development, progression and metastasis[31,32]. Traditional chemotherapeutics kill bulk tumor cells, but can not efficiently eliminate CSCs due to their overexpression of ATP-binding cassette (ABC) transporters, antiapoptotic proteins and DNA repair enzymes, resulting in drug resistance and tumor recurrence after chemotherapy[31,32].

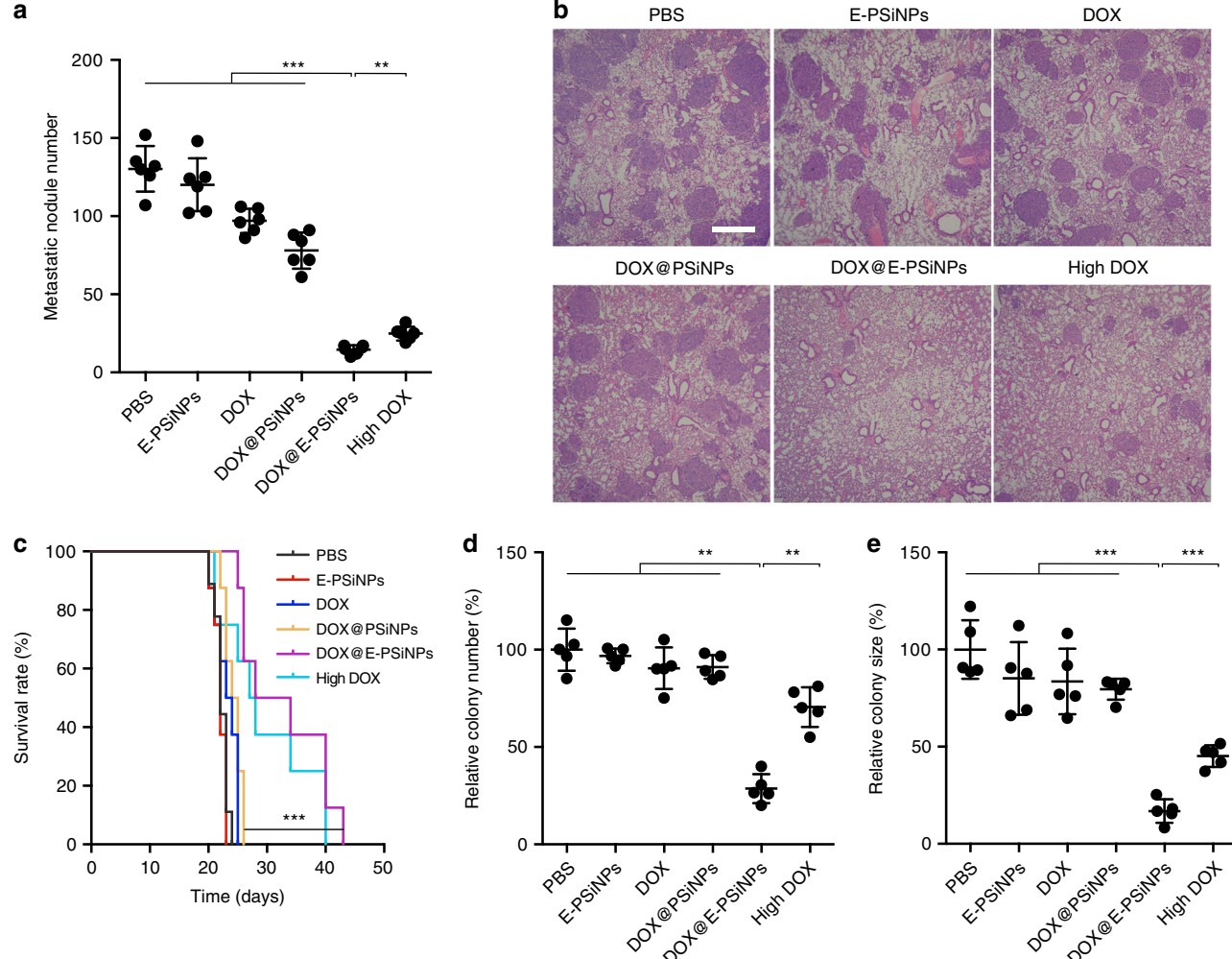

**Fig. 9** Anticancer activity of DOX@E-PSiNPs in B16-F10 lung metastasis mice. **a** Metastatic nodule numbers in lungs of B16-F10 tumor-bearing mice after intravenous injection of PBS, E-PSiNPs, free DOX, DOX@PSiNPs, DOX@E-PSiNPs exocytosed from H22 cells at DOX dosage of 0.5 mg kg$^{-1}$, or free DOX at high dosage of 4 mg kg$^{-1}$ every three days for 13 days. Data were represented as mean ± SD ($n = 6$). **b** H&E staining of lungs of B16-F10 tumor-bearing mice at the end of tumor growth inhibition experiments. Scale bar: 1000 μm. **c** Kaplan–Meier survival plot of B16-F10 tumor-bearing mice after intravenous administration of different formulations ($n = 8$). **d**, **e** Relative colony number (**d**) and size (**e**) of tumor spheroids when tumor cells digested from lung tumor nodules at the end of tumor growth inhibition experiments were seeded in soft 3D fibrin gels for 5 days. Data were represented as mean ± SD ($n = 5$). **\*\***$P < 0.01$, **\*\*\***$P < 0.001$ (one-way ANOVA with Bonferroni's multiple comparisons test for **a**, **d**, **e** and log-rank test for **c**). Source data are provided as a Source Data file

Therefore, developing effective therapeutic strategies targeted to CSCs remains a big challenge for cancer therapy.

Nowadays, some NDDSs have been successfully applied to target CSCs to treat tumor. These approaches mainly include: (1) NDDSs were rationally designed to bypass the efflux pump via endocytosis, resulting in higher intracellular accumulation in CSCs[46,47]; (2) NDDSs codelivered MDR modulators and anticancer drugs to CSCs to overcome drug resistance[48]; and (3) NDDSs were modified with CSCs targeting ligands, such as CD44[49], CD133[50], and CD90[51] to increase specificity and cellular uptake. Although these NDDSs have shown potentials to overcome chemoresistance and enhance the accumulation of anticancer drug in CSCs, they cannot achieve full therapeutic efficacy. The main reasons lie in: (1) Ideal NDDSs targeting CSCs should be characterized by enhanced tumor accumulation, tumor penetration and cellular uptake by CSCs to highly enriched in CSCs following systemic administration[52]. However, the above approaches used to target CSCs are difficult to meet all demands at the same time, hindering the therapeutic efficacy; (2) There is no universal marker used for CSCs targeting in all cancers since the markers of CSCs differ from one type of tumor to another, and these markers are often expressed by other cell types, such as normal stem cells[6,7]. Thus, targeting NDDSs to CSCs using these markers is unreliable and risky; and (3) The constructed nanoparticles usually need complicated synthesis, and are usually toxic and may cause side effects as foreign components[53]. In the present study, we developed an exosome-sheathed PSiNPs to load DOX for efficient CSCs targeting and killing. DOX@E-PSiNPs not only exhibited enhanced tumor accumulation and penetration, but also had strong cross-reactive cellular uptake and cytotoxicity against CSCs, as evidenced by the fact that DOX@E-PSiNPs exocytosed from both H22 and B16-F10 cells are efficiently internalized into H22 and B16-F10 CSCs, resulting in the strongest cytotoxicity compared with free DOX and DOX@PSiNPs. The strong cross-reactive cellular uptake of DOX@E-PSiNPs can overcome the obstacles of requiring the specific markers for targeting CSCs in different tumors. Furthermore, DOX@E-PSiNPs significantly decreased P-gp expression in CSCs, enhancing DOX retention in CSCs to overcome drug resistance. Therefore, DOX@E-PSiNPs efficiently integrated all features to eradicate CSCs, generating remarkable anticancer and CSCs killing activity in H22 tumor-bearing BALB/c mice, othotopic 4T1 tumor-bearing mice and B16-F10 tumor-bearing C57BL/6 mice. No significant toxicity of DOX@E-PSiNPs was observed in tumor-bearing mice by serological and histopathological analysis. Moreover, DOX@E-PSiNPs exocytosed from H22 cells, which were originated from liver cancer ascites of BALB/c mice, did not induce immune response in C57BL/6 mice, suggesting that DOX@E-PSiNPs are biocompatible and safe.

Upon autophagy induction, cytoplasmic materials are sequestered in double-membrane vesicles termed autophagosomes, which can fuse with MVBs to form amphisomes or directly deliver to the lysosomes for degradation[35]. Thus, the induction of autophagy usually inhibits the release of exosomes[54]. However, when the cells can not degrade material in the lysosomes due to the lysosomal defect, lysosomal overload or transport interference, the contents of lysosomes, MVBs or amphisomes are exocytosed as exosomes when fusing with cell membrane[53]. Several nanoparticles, such as silver nanoparticles[55], carbon-based nanoparticles[56] or silicon-based nanoparticles[57], were reported to induce autophagy. In this work, E-PSiNPs used as an anticancer drug carrier, were exocytosed from cancer cells in an autophagy-dependent manner. The possible reason is due to the unique structure of PSiNPs, which cannot be degraded under lysosomal acidic microenvironment[58] (Supplementary Fig. 39),

promoting cancer cells to release exosome-coated PSiNPs. The exocytosed E-PSiNPs might keep the protein integrity on exosome membranes, which can display fully the biological function of exosomes during drug delivery.

In summary, we have successfully developed biocompatible exosome-sheathed PSiNPs for targeted cancer chemotherapy. DOX@E-PSiNPs are exocytosed from tumor cells after incubation with DOX@PSiNPs. Following intravenous injection, DOX@E-PSiNPs exhibit enhanced tumor accumulation, tumor penetration and cross-reactive cellular uptake by bulk cancer cells and CSCs, resulting in augmented in vivo DOX enrichment in total tumor cells and side population cells. DOX@E-PSiNPs further demonstrate significant cross-reactive anticancer and CSCs killing activity in both subcutaneous transplantation tumor models, orthotopic tumor models and the advanced metastatic tumor models. Our study clearly demonstrates that exosome-biomimetic nanoparticles have potential as drug carriers to improve the anticancer efficacy.

## Methods

**Materials.** Boron-doped p-type silicon wafers (0.8–1.2 mΩ cm resistivity, ⟨100⟩ orientation) were produced from Virginia Semiconductor, Inc. (Fredericksburg, VA, USA). Doxorubicin hydrochloride (DOX·HCl, with purity > 98.0%) was obtained from Beijing HuaFeng United Technology CO., Ltd. (Beijing, China). RPMI 1640 medium, Dulbecco's Modified Eagle's Medium (DMEM), FBS, penicillin and streptomycin were provided by Gibco BRL/Life Technologies (Grand Island, NY, USA). Fibrinogen and thrombin were purchased from Searun Holdings Company (Freeport, ME, USA). Collagenase type I was purchased from Thermo Fisher Scientific (Waltham, MA, USA). Dispase II and TUNEL assay kit were purchased from F. Hoffmann-La Roche Ltd (Basel, Switzerland). Anti ICAM-1 antibody and anti P-gp antibody were purchased from ProteinTech (Wuhan, China). DIO, ionomycin, Hoechst 33342 and BCA protein quantification kit were purchased from Beyotime Biotechnology (Shanghai, China). Verapamil was provided by Selleck Chemicals (Houston, TX, USA). Cell counting kit (CCK-8) assay was obtained from Biosharp Company (Shanghai, China). DMA was purchased from Sigma-Aldrich (St Louis, MO, USA). All other reagents were of analytical grade and used without any further purification.

**Cell lines and animals.** Murine hepatocarcinoma cell line H22, mouse breast cancer cell line 4T1 and human hepatocarcinoma cell line Bel7402 were obtained from Type Culture Collection of Chinese Academy of Sciences (Shanghai, China). Murine melanoma cell line B16-F10 was kindly provided by Dr. Bo Huang (Huazhong University of Science and Technology, Wuhan, China). Wild type MEFs and Atg7$^{-/-}$ MEFs were kindly provided by Dr. Mingzhou Chen (Wuhan University, Wuhan, China). H22 cells were cultured in RPMI 1640 medium, and Bel7402 cells, wild type and Atg7$^{-/-}$ MEFs and B16-F10 cells were cultured in DMEM medium at 37 °C in a 5% $CO_2$ humidified incubator. All media contained 10% FBS, 100 U mL$^{-1}$ penicillin and 100 μg mL$^{-1}$ streptomycin. Six- to eight-week-old BALB/c mice (male and female) and C57BL/6 mice (male) were purchased from Beijing Vital River Laboratory Animal Technology Co., Ltd. (Beijing, China). H22 tumor-bearing mice were constructed by subcutaneously injecting 10$^6$ H22 cells per mouse into the flanks of male BALB/c mice. Orthotopic 4T1 breast tumor mode was constructed by injecting 2 × 10$^5$ 4T1 cells to the right mammary fat pad of female BALB/c mice. B16-F10 lung metastasis tumor model was constructed by intravenously injecting 5 × 10$^5$ B16-F10 cells per mouse into C57BL/6 mice. All animal experiments comply with relevant ethical regulations for animal testing and research, and were approved by the Institutional Animal Care and Use Committee at Tongji Medical College, Huazhong University of Science and Technology (Wuhan, China). All cell lines were routinely tested for mycoplasma infection and were found to be negative by MycAway-Color one-step mycoplasma detection kit.

**CSC culture.** CSCs were selected by soft 3D fibrin gels[42,43]. Fibrinogen was diluted to 2 mg mL$^{-1}$ with T7 buffer (50 mM Tris, pH 7.4, 150 mM NaCl) and then fibrinogen/cell mixtures were obtained by blending 2 mg mL$^{-1}$ fibrinogen with similar volume of cells solution (2 × 10$^3$ cells per mL), which produced 90 Pa in elastic stiffness. 250 μL mixtures were loaded into each well of 24-well plate pre-added with 5 μL thrombin (0.1 U μL$^{-1}$). The cell culture plate was then incubated at 37 °C for 30 min. Finally, 1 mL RPMI 1640 medium containing 10% FBS and antibiotics were added. On the fifth day, tumor spheroids were obtained and digested into single cells using 0.08% collagenase type I and 0.4% dispase II for 20 min at 37 °C.

**Preparation of PSiNPs and DOX@PSiNPs.** PSiNPs were prepared by electrochemical etching method[25–28]. Briefly, boron-doped p-type silicon wafers were

immersed into an aqueous solution of hydrofluoric acid (HF) and ethanol (4:1, v/v) in a Teflon etch cell, and then subjected to etch at a constant current density of 165 mA cm$^{-2}$ for 300 s. The rufous porous silicon film on the substrate was removed in 3.3% aqueous HF solution in ethanol at a constant current of 4.5 mA cm$^{-2}$ for 90 s, fragmented in ultrapure water by ultrasonication overnight, and then centrifuged at 10,000 g for 20 min to collect PSiNPs. Finally, PSiNPs were heated at 60 °C for 3 h to activate photoluminescence.

DOX@PSiNPs were prepared by adding PSiNPs in DOX solution at a weight ratio of 10:3 and then stirring for 12 h at room temperature. The mixtures were centrifuged at 10,000 g for 10 min to collect DOX@PSiNPs, followed by gently washing with ultrapure water twice to eliminate free DOX.

**Autophagy induced by PSiNPs.** H22 or Bel7402 cells were treated with 200 μg mL$^{-1}$ PSiNPs for 6 h. After washing with PBS for three times, cells were lysed in RIPA lysis buffer and then subjected to western blot analysis. Briefly, 200 μg of lysates were separated by sodium dodecyl sulfate-polyacrylamide gel electrophoresis (SDS-PAGE, 15% gel) and transferred onto nitrocellulose membranes. The membranes were blocked by 5% BSA for 2 h, and then incubated with anti-LC3 (Novus, NB100-2331SS) and anti-β-actin antibody (Beyotime, AA128, diluted to 1:2,000) at 4 °C overnight. After washing with Tris-buffered saline containing 0.1% Tween-20 (TBST), the membranes were incubated with horseradish peroxidase (HRP)-labeled secondary antibody (Beyotime, A0216, A0208, diluted to 1:10,000) at 37 °C for 2 h. The protein bands were detected using enhanced chemiluminescence (ECL) reagent and analyzed on ChemiDoc XRS Gel image system (Bio-Rad, Hercules, CA, USA). Uncropped gel images are provided in Source Data file.

Bel7402 cells were transfected with EGFP-LC3 plasmid by electroporation. After 24 h transfection, the cells were treated with 200 μg mL$^{-1}$ PSiNPs for 6 h, washed with PBS for three times and then fixed with 4% paraformaldehyde. Green fluorescence of LC3 proteins were visualized by FV1000 confocal microscope (Olympus, Japan).

**Preparation and characterization of DOX@E-PSiNPs.** To prepare E-PSiNPs or DOX@E-PSiNPs, 5 × 10$^7$ H22, Bel7402, or B16-F10 cells were treated with PSiNPs (at silicon concentration of 200 μg mL$^{-1}$) or DOX@PSiNPs (at DOX concentration of 10 μg mL$^{-1}$) for 6 h in 10 cm dishes. Subsequently, the media were discarded and replaced with fresh one without PSiNPs or DOX@PSiNPs. After 16 h incubation, the debris was discarded at 5,000 g for 15 min and then the supernatants were further centrifuged at 20,000 g for 30 min to pellet out E-PSiNPs or DOX@E-PSiNPs. Then, the obtained pellets were washed with PBS and resuspended in PBS for further experiments. DOX loading into E-PSiNPs was confirmed by labeling DOX@E-PSiNPs with DiO and then observed by FV1000 confocal microscopy. The DiO fluorescence was detected at the excitation wavelength of 488 nm and the emission range of 500–520 nm, DOX at the excitation wavelength of 559 nm and the emission range of 570–600 nm, and PSiNPs at the excitation wavelength of 488 and the emission range of 670–690 nm. The hydrodynamic diameter of E-PSiNPs and DOX@E-PSiNPs was determined by DLS (ZetaSizer ZS90, Malvern Instruments Ltd., Worcestershire, UK). The morphology of E-PSiNPs was observed by TEM (Tecnai G2-20, FEI Corp., Netherlands). DOX content loaded into DOX@E-PSiNPs was determined by incubating in 1 M NaOH for 30 min to dissolve E-PSiNPs, neutralizing with equal volume of 1 M HCl and then detecting DOX content by HPLC.

**Exosome purification.** Exosomes were purified using differential ultracentrifugation method[37,38]. First, FBS used for cell incubation was centrifuged at 100,000 g overnight to wipe out the existing exosomes. H22 or Bel7402 cells were incubated in exosome-free RPMI 1640 or DMEM medium for 48 h. Cell culture medium was collected and sequentially centrifuged at 1000 for 10 min, 10,000 g for 30 min and 100,000 g for 1 h to pellet exosomes. Exosomes were washed with PBS and recovered by centrifugation at 100,000 g for 1 h.

**Confirmation of exosomes sheathed on PSiNPs in E-PSiNPs.** E-PSiNPs were stained with 10 μM DiO for 30 min, centrifuged at 20,000 g for 30 min and then washed with PBS three times. The colocalization of DiO and PSiNPs was observed by FV1000 confocal microscopy. The fluorescence of DiO at 500–520 nm and PSiNPs at 670–690 nm was detected at the excitation of 488 nm.

E-PSiNPs was blocked by 5% BSA for 30 min, and then incubated with FITC-conjugated CD63 antibody (Biolegend, 353005, diluted to 1:200) for 30 min at room temperature. The colocalization of CD63 and PSiNPs was observed by FV1000 confocal microscopy. The fluorescence of FITC at 500–520 nm and PSiNPs at 670–690 nm was detected at the excitation of 488 nm.

Whole cells, the purified exosomes and E-PSiNPs were lysed in RIPA lysis buffer and then subjected to western blot analysis. The primary antibodies used included anti-CD63 (Abcam, ab216130), anti-TSG101 (Santa Cruz, SC-7964) and anti-calnexin (Beyotime, AC018). All primary antibodies were diluted to 1:2000. Uncropped gel images are provided in Source Data file.

**E-PSiNPs yield.** Bel7402 cells were incubated with 200 μg mL$^{-1}$ PSiNPs for 6 h and washed with PBS three times. Then fresh medium containing 200 nM

rapamycin, 30 μM CBZ, 5 mM 3-MA, 15 nM DMA or 10 μM ionomycin were added. After 16 h incubation, the supernatants were collected, centrifuged at 5000 g for 15 min to remove debris, and then centrifuged at 20,000 g for 30 min. The pellets were dissolved in 1 M NaOH solution and silicon content was measured by Optima 4300 DV ICP-OES (PerkinElmer, Norwalk, CT, USA).

**In vitro DOX release profile.** DOX release profile from DOX@E-PSiNPs was determined by dialysis method. Briefly, DOX@E-PSiNPs (300 μg DOX content) were put into a dialysis bag (cutoff molecular weight was 3000 Da) and submerged fully into PBS (30 mL), then stirred with 250 rpm at 37 °C. At the designated time intervals, 0.5 mL of sample solution was taken out and replaced with equal amount of fresh PBS. DOX content in samples was measured by HPLC.

**Interaction between DOX@E-PSiNPs and CSCs by AFM.** H22 CSCs were seeded on coverslips pretreated with poly-lysine in 6-well plates at a density of 3 × 10$^5$ cells per well. H22 CSCs were then incubated with DOX, DOX@PSiNPs or DOX@E-PSiNPs at the DOX concentration of 2 μg mL$^{-1}$ at 37 °C for 2 h. After washing with PBS, CSCs were fixed with 0.25% (v/v) glutaraldehyde for 30 min at room temperature. The coverslips were rinsed with deionized water to remove salt crystals and air dried before analysis. AFM images were obtained using a multi-mode 8 AFM (Bruker, Santa Barbara, CA, USA). Cell surface studies were performed in ScanAsys mode at scan frequencies below 1 Hz. The roughness of CSCs membrane was analyzed by measurement of Image Rq.

**Cell membrane fluidity.** H22 CSCs membrane fluidity was measured using fluorescence polarization of 1,6-diphenyl-1,3,5-hexatriene (DPH)[59]. Briefly, H22 CSCs (10$^6$ cell per mL) were incubated with DPH (2 μM) at 37 °C for 1 h. The labeled CSCs were then incubated with DOX, DOX@PSiNPs or DOX@E-PSiNPs at the DOX concentration of 2 μg mL$^{-1}$ for 2 h. Fluorescence anisotropy was measured using a polarization spectrofluorometer (FP-6500, Jasco, Tokyo, Japan) with an excitation wavelength of 365 nm and an emission wavelength of 429 nm. Anisotropy was calculated as: $r = (I_{VV} - I_{VH}.G)/(I_{VV} + I_{VH}.G)$, where $G = I_{HV}/I_{HH}$ is used to correct the unequal transmission of the optics.

**Internalization into bulk cancer cells and CSCs.** Bel7402 and H22 cells, and H22 and B16-F10 CSCs selected in soft 3D fibrin gels were seeded into six-well plates overnight at a density of 2 × 10$^5$ cells per well. Subsequently, cells were incubated with free DOX, DOX@PSiNPs or DOX@E-PSiNPs at different DOX concentrations for 2 h, rinsed with PBS and then collected to analyze the intracellular DOX fluorescence in FL2 channel by flow cytometry (FC500, Beckman Coulter, Fullerton, CA, USA).

**Cytotoxicity against bulk cancer cells and CSCs.** For determination of DOX@E-PSiNPs against bulk cancer cells, Bel7402, H22 and B16-F10 cells were seeded in 96-well plate at a density of 8 × 10$^3$ cells per well overnight and then treated with free DOX, DOX@PSiNPs or DOX@E-PSiNPs at different DOX concentrations. After 24 h treatment, cell survival rate was detected by CCK-8 assay.

To evaluate cytotoxicity of DOX@E-PSiNPs against CSCs, H22 and B16-F10 cells were pre-treated with DOX, DOX@PSiNPs or DOX@E-PSiNPs at different DOX concentrations for 4 h. The cells were harvested, washed and counted. 4 × 10$^2$ cells from different groups were then seeded in 3D fibrin gels. On the fifth day, the numbers of tumor spheroids in different groups were counted under Olympus IX 71 optical microscope (Tokyo, Japan). Tumor spheroids in each group were imaged and their sizes were calculated by Image J software.

To further determine cytotoxicity of DOX@E-PSiNPs against CSCs, H22 and B16-F10 CSCs were selected in 3D fibrin gels in 96-well plates. On day 5, the media were aspirated and fresh media containing DOX, DOX@PSiNPs or DOX@E-PSiNPs at DOX concentration of 2 μg mL$^{-1}$ were added. After 24 h incubation, tumor spheroids with integral rims in each group were counted under optical microscope. The images of tumor spheroids in each group were captured and their sizes were calculated by Image J software.

**In vivo biodistribution.** When tumor volume of H22 tumor-bearing mice reached ca. 250 mm$^3$, or at 13 days after intravenous injection of B16-F10 cells into C57BL/6 mice, the mice were intravenously injected with free DOX, DOX@PSiNPs or DOX@E-PSiNPs exocytosed from H22 cells at DOX dosage of 0.5 mg kg$^{-1}$, or free DOX at high dosage of 4 mg kg$^{-1}$. At 24 h post-injection, the mice were sacrificed, and the major organs (heart, liver, spleen, lung and kidney) and tumors in H22 tumor-bearing mice and lung metastatic nodules in B16-F10 tumor-bearing mice were collected. Subsequently, tissues were lysed and DOX was extracted by incubating the lysates in 1 M NaOH for 30 min and neutralizing by the same volume of 1 M HCl. DOX contents in the lysates were tested by a FlexStation3 Multi-Mode Microplate Reader (Molecular Devices, Sunnyvale, CA, USA).

**In vitro penetration in 3D tumor spheroids.** H22 tumor spheroids were constructed using soft 3D fibrin gel method as described above. To observe penetration of DOX@E-PSiNPs from outside of tumor spheroids to core area, tumor spheroids with diameter of ca. 150–200 μm were treated with free DOX, DOX@PSiNPs or

DOX@E-PSiNPs exocytosed from H22 cells at DOX concentration of 2 µg mL$^{-1}$ for 24 h. The spheroids were washed with PBS twice, fixed with 4% paraformaldehyde for 30 min, and then transferred to confocal dishes. DOX red fluorescence at 570−590 nm was observed by confocal microscopy using Z-stack scanning mode at the intervals of 5 µm at the excitation of 559 nm. The 3D DOX fluorescence images in tumor spheroids were reconstructed by using Amira software.

**In vivo tumor penetration**. When tumor volume of H22 tumor-bearing mice reached ca. 250 mm$^3$, the mice were intravenously injected with DOX, DOX@P-SiNPs or DOX@E-PSiNPs exocytosed from H22 cells at DOX dosage of 0.5 mg kg$^{-1}$. At 24 h post-injection, tumor tissues were collected, washed with PBS, and then frozen-sectioned into pieces. The sections were incubated with FITC-CD31 antibody (Biolegend, 102405, diluted to 1:200) at 37 °C for 30 min to label tumor vessels, and then rinsed with PBS. DOX red fluorescence at 570−600 nm and FITC-CD31 green fluorescence at 500−520 nm were observed by confocal microscopy at the excitation of 559 nm or 488 nm, respectively. DOX distribution from blood vessels to deep tumor tissues was measured by Image J Software.

**Intercellular delivery**. H22 cells were seeded on coverslips, which were pretreated with 10 µg mL$^{-1}$ poly-lysine overnight. Cells on the first coverslip were treated with 2 µg mL$^{-1}$ DOX, DOX@PSiNPs or DOX@E-PSiNPs for 6 h. The treated cells were rinsed with PBS and then co-incubated with the new cells on the second coverslip for 16 h in fresh medium. Finally, the cells on the second coverslip were co-incubated with the new cells on the third coverslip for another 16 h in fresh medium. Cells were rinsed with PBS and the intercellular DOX fluorescence at 570–600 nm were analyzed by confocal microscope at the excitation of 559 nm and flow cytometry (FL2 channel).

**In vivo DOX accumulation in total tumor cells and CSCs**. When tumor volume of GFP-expressing H22 tumor-bearing mice reached ca. 250 mm$^3$, the mice were intravenously injected with free DOX, DOX@PSiNPs or DOX@E-PSiNPs exocytosed from H22 cells at DOX dosage of 0.5 mg kg$^{-1}$, or free DOX at high dosage of 4 mg kg$^{-1}$. At 24 h after injection, tumor tissues were collected, washed with PBS and then cut into small pieces, followed by digestion with 1 mg mL$^{-1}$ collagenase type I solution at 37 °C for 2 h. The single tumor cells were acquired by filtering the digested cells with 200-mesh nylon twice and then divided into two parts. One part was used to determine DOX content in total GFP-positive tumor cells by flow cytometry in FL2 channel. The other one was applied to determine DOX content in side population cells of GFP-positive tumor cells. The tumor cells were treated with 5 µg mL$^{-1}$ Hoechst 33342 for 90 min in the presence or absence of 50 µm verapamil at 37 °C in the dark. Living cells were plotted on SSC-A and FSC-A graph. Hoechst 33342 fluorescence at 450 nm or 660 nm in living GFP-positive tumor cells was measured. The gating of side population cells was plotted as the absence of cell population in PBS-treated group comparted with verapamil-treated group. DOX fluorescence intensity in side population cells was detected by flow cytometry (FL2 channel).

**Anticancer activity in subcutaneous H22 tumor-bearing mice**. When tumor volume of H22 tumor-bearing mice reached ca. 200 mm$^3$, the mice were intravenously injected with PBS, E-PSiNPs, free DOX, DOX@PSiNPs, DOX@E-PSiNPs exocytosed from H22 cells at DOX dosage of 0.5 mg kg$^{-1}$, or free DOX at high dosage of 4 mg kg$^{-1}$ once every three days ($n = 14$ per group). The tumor sizes were measured every day via vernier caliper and the body weights of mice were also recorded. On 17th day of treatment, mice were further divided into two groups. One group ($n = 8$) was used for survival experiment, while the other part ($n = 6$) was used to estimate anticancer efficacy. For anticancer efficacy analysis, mice were sacrificed, and tumors and major organs (heart, liver, spleen, lung and kidney) were obtained and washed with PBS. The cleaned tumors were weighed and fixed with 4% paraformaldehyde, then sectioned and stained using a TUNEL assay kit according to the manufacturer's protocol. The major organs were also fixed with 4% paraformaldehyde, sectioned and examined by H&E staining.

To estimate CSCs killing activity of DOX@E-PSiNPs in vivo, tumor tissues were digested into single cells after treatment as above using 1 mg mL$^{-1}$ collagenase type I solution. One part of cells were used to determine the number of CD133-positive cells by flow cytometry. The second part of cells were seeded in soft 3D fibrin gels (400 cells per well) and incubated for 5 days[42,43]. The numbers of tumor spheroids were counted under optical microscope. The images of tumor spheroids were captured and their sizes were calculated by Image J software. The third part of cells were subcutaneously transplanted into BALB/c mice (10$^6$ cells per mouse) and the tumor formation ratio was evaluated. Furthermore, the anticancer treatment was performed in the GFP-expressing H22-tumor bearing mice as above. On 17th day of treatment, the mice were sacrificed and tumor tissues were digested into single cells. The cells were dyed with 5 µg mL$^{-1}$ Hoechst 33342 for 90 min at 37 °C in the presence or absence of 50 µM verapamil in the dark, then washed and resuspended in PBS. The number of side population cells in GFP-positive tumor cells was measured by flow cytometry (450/45 BP filter for blue fluorescence and 660/20 BP filter for red fluorescence).

**Anticancer activity in orthotopic 4T1 breast cancer mode**. 4T1 cells ($2 \times 10^5$ cells) were suspended in 50 µL PBS and then injected into the right forth breast fat pad. When the tumor volume reached 50−70 mm$^3$, the mice were administrated with PBS, E-PSiNPs, free DOX, DOX@PSiNPs or DOX@E-PSiNPs at DOX dosage of 0.5 mg kg$^{-1}$, or high dosage of DOX at 4 mg kg$^{-1}$ once every three days for five times ($n = 14$ per group). The tumor sizes were measured every day via vernier caliper. On 15th day of treatment, mice were further divided into two groups. One group ($n = 8$) was used for survival experiment, while the other part ($n = 6$) was used to estimate anticancer efficacy.

To investigate CSCs killing activity of DOX@E-PSiNPs in vivo, tumor tissues were collected and digested into single cells using 1 mg mL$^{-1}$ collagenase type I solution. The cells (400 tumor cells per well) were seeded in soft 3D fibrin gels. On day 5, the numbers of tumor spheroids were counted under optical microscope. The images of tumor spheroids were captured and their sizes were calculated by Image J software.

**Anticancer activity in B16-F10 lung metastasis cancer model**. At 48 h after B16-F10 cells ($5 \times 10^5$ cells per mouse) were intravenously injected into C57BL/6 mice, the mice were intravenously administrated with PBS, E-PSiNPs, free DOX, DOX@PSiNPs, DOX@E-PSiNPs exocytosed from H22 cells at DOX dosage of 0.5 mg kg$^{-1}$, or free DOX at high dosage of 4 mg kg$^{-1}$ once every three days ($n = 14$ per group). On 13th day of treatment, mice were divided into two parts. One part (n = 8) was used for long-term survival experiment and the other part ($n = 6$) was used for evaluation of anticancer effect. For evaluation of anticancer effect, the mice were sacrificed and the lungs were acquired. The numbers of tumor nodules on the surface of lungs were recorded. Lungs were then fixed with 4% paraformaldehyde, sectioned and examined by H&E staining.

To investigate CSCs killing activity of DOX@E-PSiNPs in vivo, tumor nodules were collected and digested into single cells using 1 mg mL$^{-1}$ collagenase type I solution. The cells (400 tumor cells per well) were seeded in soft 3D fibrin gels. On day 5, the numbers of tumor spheroids were counted under optical microscope. The images of tumor spheroids were captured and their sizes were calculated by Image J software.

**Immune response**. C57BL/6 mice were intravenously injected with PBS, E-PSiNPs, DOX, DOX@PSiNPs, DOX@E-PSiNPs exocytosed from H22 cells at the DOX dosage of 0.5 mg kg$^{-1}$, or free DOX at high dosage of 4 mg kg$^{-1}$. At different time intervals, the orbital blood was obtained, maintained for 30 min and centrifuged at 10,000 g for 10 min. The serum was collected and the contents of IgM, IL-1β, IL-6, and TNF-α were analyzed by enzyme linked immunosorbent assay (ELISA).

**Statistical analysis**. Experiments were performed with at least three replicates. All values were presented as mean values ± SD. Statistical analyses were carried out using the GraphPad Prism software version 6.0. Comparison between two groups was performed using unpaired two-tailed Student's $t$ test. One-way ANOVA or two-way ANOVA was used for comparison of more than two groups. Statistical significance for survival curves was determined using a log-rank test. Values with $P < 0.05$ are considered significant.

**Reporting summary**. Further information on research design is available in the Nature Research Reporting Summary linked to this article.

## Data availability
The authors declare that the main data supporting the findings of this study are available within the article and its Supplementary Information. Extra data are available from the corresponding author upon reasonable request. The source data underlying Figs. 2–9 and Supplementary Figs. 1–39 are provided with the paper as a Source Data file. A reporting summary for this article is available as a Supplementary Information file.

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

## Acknowledgements

This work was supported by National Basic Research Program of China (2018YFA0208900 and 2015CB931800), National Natural Science Foundation of China (81627901, 81672937, 81773653 and 81803018), Program for HUST Academic Frontier Youth Team (2018QYTD01), Program for Changjiang Scholars and Innovative Research Team in University (IRT13016), Academy of Finland (297580), Sigrid Jusélius Foundation (28001830K1 and 4704580), HiLIFE Research Funds and the European Research Council proof-of-concept grant (decision no. 825020). We thank the Research Core Facilities for Life Science (HUST), the Analytical and Testing Center of Huazhong University of Science and Technology and Wuhan institute of biotechnology for related analysis.

## Author contributions

L.G., H.A.S., and X.Y. designed the project. T.Y., Xiaoqiong Z., N.B., H.Z., Xuting Z., F.L., A.H., and J.H. performed the experiments. T.Y., Xiaoqiong Z., N.B., H.Z., L.G., H.A.S. and X.Y. analyzed and interpreted the data, and wrote the paper

## Additional information

**Competing interests:** The authors declare no competing interests.

