## [Peer Review File · Nature Communications]

Reviewers' comments:

Reviewer #1 (Expertise: Biomimetic nanoparticle delivery, Remarks to the Author):

In this manuscript, the authors report on an exosome-coated porous silicon nanoparticle platform for cancer drug delivery. The particles are fabricated by incubating bare silicon nanoparticles with live cells, which then package the particles within membranes before exocytosing them. The resultant particles express characteristic exosome markers and can also be produced at relatively high yield. Interestingly, the particles are capable of targeting different types of cancer cells, and exhibit enhanced efficacy when loaded with doxorubicin, as compared with free drug alone. The authors conclude with in vivo efficacy studies demonstrating that their platform can effectively control tumor growth in both primary tumor and metastasis models. Overall, the idea of exosome membrane coating to enable targeted delivery is interesting. However, there are also a significant number of technical issues that need to be addressed.

(1) While the idea of cross-reactivity of the exosome nanoparticles is interesting, the authors need to definitively demonstrate this idea with the appropriate controls. For example, they should test if the reverse scenario (B16 exosomes to H22 cells) gives a similar result. They should also test either the targeting of healthy cells or use exosome nanoparticles derived from healthy cells. Furthermore, the cross-reactivity of their platform is not explained very well. Why can the particles secreted from one cell type be used to target multiple cell types? Can the authors identify specific markers that are responsible for this effect?

(2) The mechanism of particle formation via exocytosis would be more convincing if the authors could visualize what is depicted in Scheme 1A by TEM of histological cell sections. Is the property of autophagy induction unique to the porous silicon nanoparticles or is it universal? If unique, what type of nanoparticle properties are desirable for inducing autophagy?

(3) In the western blot study with the exosomes purified by ultracentrifugation, the authors don't show any enrichment of CD63, which indicates that their purification process was not optimized. Further, they showed enrichment of Tsg101, which is not a membrane-bound protein, in their coated nanoparticle formulation. How do they explain these results?

(4) For the nanoparticle characterization, why was the number distribution used instead of intensity distribution. The Z-average sizes of both the uncoated and coated particles, along with the PDI of each, should be reported. How can the authors explain that the size distribution of the coated

particles is better than the uncoated particles? How do the authors explain the large size increase on DLS, whereas their TEM image shows minimal size increase?

(5) The fluorescence of DiO/FITC and DOX are not well separated. It is surprising that the authors were able to resolve these fluorophores. Are the authors certain that their imaging data using these dyes are reliable?

(6) How do the authors explain the poor drug loading efficiency? Loading at <1% efficiency is not economical for future translation. How did the authors ensure that their uncoated and coated formulations had equivalent drug loading before testing them in vitro and in vivo? It seems unlikely that this process can be controlled well enough to achieve equivalent loading given the complexity of the coating process. How did the authors ensure that they released all of the DOX from their nanoparticles when applying their HPLC protocol?

(7) Their formulation is fairly large (almost 300 nm). It seems unlikely that such a large nanoformulation can penetrate tumors by traveling through the interstitial space. Is there another mechanism to explain their deep penetration?

Minor issues:

- The description of cancer stem cells as “side population cells” is confusing. Is this a common name for them?
- The authors may be overstating their claim that traditional methods for coating membrane onto nanoparticles are disruptive.
- Can the authors provide the absorbance profile and the emission profile at the appropriate excitation wavelength of their porous silicon nanoparticles?
- Can the authors demonstrate the colocalization of their nanoparticles with LC3 in Figure 1B?
- For their survival curves, what was the exact statistical test that was applied? If by the log-rank test, it seems highly unlikely that the difference between their last two groups was significant for either experiment. Can they provide the actual p-values?

Reviewer #2 (Expertise: Silicon based NP, therapy, Remarks to the Author):

This is a well-done, multi-faceted study involving an innovative approach to targeted anticancer therapy using an exosome sheath to provide recognition for a DOX-loaded porous silicon drug delivery platform. The authors describe outcomes from a rational series of experiments – demonstration of efficient cellular uptake by cancer cells/CSCs, along with strong cytotoxicity and enhanced tumor accumulation and penetration - whose conclusions build iteratively in a logical manner. This is a significant advance in retention of surface protein selectivity and (hopefully) stability in nanoscale drug delivery in useful anti-cancer therapies.

I very much enjoyed reading and analyzing this manuscript. I do have some focused questions that I would like the authors to address before publication can be recommended:

(1) The porous Si platform is a key component in controlled, sustained delivery. A key question that requires additional commentary in this manuscript is both the morphology/microstructure of the initial porous Si particles (TEM images + size distribution data) along with evolution of the porous Si morphology, if any, before/after incorporation into the exosome construct? [hydrodynamic radii data are provided, but is there any change in morphology of individual porous Si particles (on a statistical basis, observed via SEM/TEM?). OES ICP measured total dissolved Si content, but does not provide useful structural data. A second related question concerns any evidence (or lack thereof) in porous Si particle morphology during the measured therapeutic windows utilized in these experiments?

(2) Porous Si nanoparticle degradation *in vitro* / *in vivo* is quite sensitive to feature size and surface chemistry. It can be quite fast (hours) or slow (weeks to months), depending on the above factors. Neither of these key pieces of information are readily noted in the manuscript in its present form, only that the particles are unchanged after 6 days exposure *in vitro* (page 12). This sort of detail is important to understanding the long term behavior of this material *in vivo*.

(3) Other than sustained delivery, is there evidence for any unique structural contribution of the nanoscale Si carrier? If so, these points should be emphasized here.

(4) In this regard, the authors only address three references to porous Si drug delivery platforms (unless I am mistaken), and there needs to be more added. There is no shortage of possible choices, as the authors are likely aware.

(5) Ultimate clinical relevance is dependent on multiple factors, of course, one of which is shelf-life. What is the stability of a exosome- packaged, DOX-loaded porous Si material?

I look forward to reading the authors response and revised manuscript.

Reviewer #3 (Expertise: Liver cancer models, therapy, Remarks to the Author):

This work represents physical and chemical characterization of silicon-based, exosome-encapsulated nanoparticles loaded with doxorubicin, and in vitro and preclinical in vivo proof of concept studies using hepatocellular carcinoma. The manuscript is reasonably well written but there are several errors in English grammar and spelling which should be corrected. Furthermore, from a model and therapeutic point-of-view, it would be useful for the authors to address the following:

1. The Dox@E-PSiNPs are generated from a human hepatocellular carcinoma line, but are delivered to murine species. What is the immunocompetence of the murine models and have the authors performed any immunologic assessments to determine if a cellular or humoral immune response develops? How would the authors propose avoiding an immunologic reaction in human patients treated with these exosome-encapsulated NPs?

2. Figure 5 indicates that a high level of Dox@E-PSi-NPs is discovered in non-malignant liver, actually at much higher levels than are seen in the malignant hepatic tumors. What cells in the liver are accumulating these NPs; Kupffer cells, hepatocytes, or some other cell type? Is there any liver toxicity, inflammation, or sub-acute liver injury noted in the animals treated repeatedly with these NPs?

3. Treatment of hepatocellular carcinoma or melanoma with doxorubicin has limited success clinically, primarily due to induction of P-glycoprotein with increased levels of this trans-membrane protein responsible for chemotherapy resistance following doxorubicin exposure. Have the authors evaluated P-glycoprotein expression in their cancer cell lines, including CSCs, following treatment with the doxorubicin-loaded NPs? The authors results demonstrate improved response and prolonged survival in vitro and in animals treated with the novel NPs, but no complete response is seen. Can the authors offer any data explaining the lack of any complete responders? Is this due to CSCs developing a resistant phenotype?

Reviewer #4 (Expertise: CSCs and in vivo models, Remarks to the Author):

In this study, Yong et al. developed a novel biocompatible exosome sheathed PSiNPs (ePSiNPs) as a delivery vehicle for targeted cancer chemotherapy. The authors demonstrated autophagy is involved in exocytosis of PSiNPs. Furthermore, they showed that the exocytosed PSiNPs is sheathed by exosomes. Once the PSiNPs was loaded with doxorubicin and intracellularly processed to form DOX@ePSiNPs, it shows stronger uptake efficiency and cytotoxicity for cancer stem cells (CSCs). In vivo study reveals that DOX@E-PSiNPs exhibits the highest anti-cancer effect in comparison to free doxorubicin and DOX@PSiNPs.

Major concerns:

1. Is the side population confirmed to be CSCs in these tumor cell lines? Authors should also test other verified CSC markers for these tumor models.
2. Authors showed that DOX@E-PSiNPs can shrink the tumor very fast and also decreased the CSC percentage significantly more than other three controls. But, the CSC percentage change doesn't mean the absolute CSC number change which is based on the total tumor cell number in the tumor. The golden standard to measure the CSC change is to do the secondary transplantation from the primary tumors. It looks like DOX@E-PSiNPs target both CSC and non-CSC simultaneously, if the absolute CSC number is higher in DOX@E-PSiNPs treated group, it might promote the tumor relapse and metastasis.
3. "CSCs located within the hypoxic core of tumor mass" on Page 15 is not exactly right since it's been confirmed that some metastatic CSCs are located in the edge of the tumors.
4. The claim "Significantly fewer metastatic nodules were detected in the DOX@E-PSiNPs-treated group" has to be further demonstrated that fewing metastatic nodules was due to DOX@E-PSiNPs targeting the metastatic tumor cells from the primary cells or targeting the tumor cells in the lung?
5. The use of cell lines for the study is not well designed and confusing, for example, in Fig 1 through Fig. 3, they used Bel7402 to test the involvement of autophagy and so on, somehow they skipped to H22 and B16-F10 in Fig 4 to test cellular uptake and cytotoxicity of DOX@E-PSiNPs without explanation. Similarly, they applied H22 to in vivo model to test the enrichment and CSCs killing activity of DOX@ePSiNPs, then used B16-F10 for the lung metastasis model. It's understandable that authors want to test in different tumor types, but one tumor type should be tested for all studies in the paper and other tumor types can be tested for a support.
6. For Fig 2C and 2D, Dio and anti-CD63 antibody should also be incubated with non-exocytosed PSiNPs as negative control to exclude the possibility of non-specific binding.
7. For Fig 2E, is β -actin a proper internal control for both of total cell lysates and exosomes? The authors can display a ponceau S stained membrane as the reference of loaded protein.
8. In Fig 5A, DOX@E-PSiNPs shows very high accumulation of DOX in liver, kidney and spleen (even higher than free DOX). This result indicates potential side effects of ePSiNPs as delivery vehicle

for chemotherapy. The authors should discuss how to improve the specificity of ePSiNPs for targeting tumor cells.

9. For Fig 5C, the authors claimed that DOX delivered with DOX@E-PsiNPs is detectable at deeper location, up to 400 μm from blood vessel in the text, somehow in the figure, the signal of DOX@E-PsiNPs is only detectable at the depth of 100 μm . In the contrast, DOX@PsiNPs is detectable at the depth of 400 μm .

10. For Fig 6D, according to the description of procedure, cancer cells were not isolated from stroma cells. Thus the SP cells may not represent CSCs population. To address this question, the authors can label H22 cells with GFP for injection. GFP positive cells can be gated prior to the analysis of SP cells.

Minor concerns

1. Normal cancer cells should be changed to bulk cancer cells
2. "Discussion" section is too simple or missing.
3. For Fig 1B, actual length of the scale bar was not described in figure legends.
4. For Fig 3D, statistical significance should be displayed at the last time point.
5. For Fig 5A, labels of X-axis is not well displayed (partially hidden).
6. For Fig 5B, the meaning of the white bars on pictures of DOX group was not described in figure legends.
7. In Supplementary Fig. 8, the authors didn't show the data regarding the uptake efficiency of DOX@E-PsiNPs by B16-F10, just like the results in supplementary Fig. 7 for H22 and Bel7402. Does this mean DOX@E-PsiNPs exhibit no higher intracellular internalization than free DOX and DOX@PSiNPs in B16-F10? This question should be addressed clearly.

RESPONSE TO REVIEWERS

We would like to express our sincere thanks to all reviewers for their critical and constructive comments. We have performed substantial additional experiments to address their concerns. We respond point-by-point to each of their comments and criticisms. We feel that their comments have helped us to significantly improving and strengthening the manuscript, as well as clarifying some important issues of our work. We hope that the revision has addressed their major concerns.

RESPONSE TO REVIEWER #1:

In this manuscript, the authors report on an exosome-coated porous silicon nanoparticle platform for cancer drug delivery. The particles are fabricated by incubating bare silicon nanoparticles with live cells, which then package the particles within membranes before exocytosing them. The resultant particles express characteristic exosome markers and can also be produced at relatively high yield. Interestingly, the particles are capable of targeting different types of cancer cells, and exhibit enhanced efficacy when loaded with doxorubicin, as compared with free drug alone. The authors conclude with in vivo efficacy studies demonstrating that their platform can effectively control tumor growth in both primary tumor and metastasis models. Overall, the idea of exosome membrane coating to enable targeted delivery is interesting. However, there are also a significant number of technical issues that need to be addressed.

1. While the idea of cross-reactivity of the exosome nanoparticles is interesting, the authors need to definitively demonstrate this idea with the appropriate controls. For example, they should test if the reverse scenario (B16 exosomes to H22 cells) gives a similar result. They should also test either the targeting of healthy cells or use exosome nanoparticles derived from healthy cells. Furthermore, the cross-reactivity of their platform is not explained very well. Why can the particles secreted from one cell type be used to target multiple cell types? Can the authors identify specific markers that are responsible for this effect?

Response:

We thank the reviewer's constructive suggestion. According to the reviewer's suggestion, we first determined the cellular uptake and cytotoxicity of DOX@E-PSiNPs exocytosed from B16-F10 cells against H22 CSCs. Consistent with the previous results, DOX@E-PSiNPs exocytosed from B16-F10 cells exhibited stronger cell uptake by H22 CSCs than free DOX and DOX@PSiNPs (Supplementary Fig. 13A in the revised manuscript). Correspondingly, fewer colony number and smaller colony size were observed when H22 cells pretreated with free DOX, DOX@PSiNPs or DOX@E-PSiNPs exocytosed from B16-F10 cells at different DOX concentrations for 4 h were seeded in soft 3D fibrin gels for 5 days (Supplementary Fig. 13B,C in the revised manuscript). Furthermore, when H22 CSCs selected in 3D fibrin gels for 5 days were treated with free DOX, DOX@PSiNPs or DOX@E-PSiNPs exocytosed from B16-F10 cells at the DOX concentration of 2 µg/mL for 24 h, DOX@E-PSiNPs still exhibited the strongest inhibition in colony number and

size of tumor spheroids (Supplementary Fig. 13D,E in the revised manuscript). These results further confirmed that DOX@E-PSiNPs exhibited a strong cross-reactive cellular uptake and cytotoxicity against CSCs, irrespective of their origin. Moreover, we also evaluated the cellular uptake and cytotoxicity of DOX@E-PSiNPs exocytosed from H22 cells against B16-F10 cells and DOX@E-PSiNPs exocytosed from B16-F10 cells against H22 cells. Undoubtedly, DOX@E-PSiNPs exhibited the strong cross-reactive cellular uptake and cytotoxicity against cancer cells (Supplementary Fig. 15 in the revised manuscript). We added these new data in the revised manuscript, Page 15 and Supplementary Figs. 13 and 15.

To further evaluate the tumor cell targeting capacity of tumor exosome-coated PSiNPs, we determined the cellular uptake of DOX@E-PSiNPs exocytosed from healthy human umbilical vein endothelial cells (HUVEC, DOX@E_{HUV}-PSiNPs) by H22 cells or DOX@E-PSiNPs exocytosed from H22 cells (DOX@E_{H22}-PSiNPs) by HUVEC cells. Consistent with the previous data, DOX@E_{H22}-PSiNPs exhibited the strongest cellular uptake by H22 cells compared with free DOX, DOX@PSiNPs or DOX@E_{HUV}-PSiNPs. However, the cellular uptake of DOX@E_{H22}-PSiNPs by HUVEC cells was significantly lower than that of DOX@E_{HUV}-PSiNPs, similar to that of free DOX or DOX@PSiNPs. These results showed that DOX@E_{H22}-PSiNPs exhibited strong tumor cell targeting capacity. We added these new data in the revised manuscript, Page 15 and Supplementary Fig.17.

The adhesion of molecules plays an important role in mediating intercellular and cell-extracellular matrix interactions of cancer (*Nat. Rev. Mol. Cell. Bio.* 2011, 12: 189). The membrane vesicles of any cellular origin express adhesion molecules on their surface (*Nat. Rev. Immunol.* 2014, 14: 195). We found that CD54 (also named intercellular adhesion molecule 1, ICAM1) was expressed in DOX@E-PSiNPs exocytosed from H22 and B16-F10 cells. To determine whether CD54 was involved in the cross-reactive cellular uptake of DOX@E-PSiNPs by cancer cells, H22 or B16-F10 cells were treated with DOX@E-PSiNPs exocytosed from H22 cells for 2 h which were pretreated with or without CD54 antibody for 2 h at 4 °C, followed by intracellular DOX fluorescence measurement. Pretreatment with CD54 antibody significantly decreased intracellular DOX fluorescence in DOX@E-PSiNPs-treated H22 or B16-F10 cells. These results indicated that CD54 played an important role in the regulation of cross-reactive cellular uptake of DOX@E-PSiNPs by cancer cells. We added these new data in the revised manuscript, Page 15 and Supplementary Fig. 16.

2. The mechanism of particle formation via exocytosis would be more convincing if the authors could visualize what is depicted in Scheme 1A by TEM of histological cell sections. Is the property of autophagy induction unique to the porous silicon nanoparticles or is it universal? If unique, what type of nanoparticle properties are desirable for inducing autophagy?

Response:

We appreciate the reviewer's constructive comments. According to the reviewer's suggestion, Bel7402 cells were treated with 200 µg/mL PSiNPs for different time courses (0, 2, 4, 6 or 12 h), followed by fixing in 4% formaldehyde and 1% glutaraldehyde and processing for TEM. As shown in the **Fig. R1**, obvious autophagosomes and multivesicular bodies (MVBs) were observed in cells treated with PSiNPs. However, it was hard to identify that PSiNPs were localized in autophagosomes or MVBs by Energy Dispersive Spectrometer (EDS), followed by exocytosis because of the inherent Si elements in cells. Instead, we determined the colocalization of PSiNPs with EGFP-LC3-labeled autophagosomes in the revised manuscript, suggesting intracellular PSiNPs being captured in the LC3⁺ autophagosomes (Fig. 1B in the revised manuscript). Moreover, our data in the original manuscript further showed that autophagy mediated the exocytosis of PSiNPs, as evidenced by the decreased exocytosis of PSiNPs in response to 3-MA (an autophagy inhibitor) treatment and the enhanced exocytosis of PSiNPs in response to rapamycin and CBZ (autophagy inducers). Furthermore, PSiNPs were colocalized with FITC-CD63-labeled MVBs after internalization, and the exocytosed E-PSiNPs expressed exosomes markers. Dimethyl amiloride (DMA, an inhibitor of exosome release) significantly decreased, yet ionomycin (a promoter of exosome release) significantly increased the yield of E-PSiNPs. Altogether, our data support the fact that PSiNPs-induced autophagy regulates their exocytosis after internalization and exosomes are coated with the exocytosed PSiNPs.

Autophagy is a highly regulated process for intracellular homeostasis through clearance, degradation, or exocytosis of damaged cell components or foreign risks (*Cell* 2015, 161: 1306). Several nanoparticles, including ceria, silver, carbon nanoparticle, etc., have been reported to induce autophagy (*ACS Nano* 2014, 8: 10328; *Autophagy* 2014, 10: 2006; *ACS Nano* 2014, 8: 2087). Therefore, the nanoparticles-induced autophagy might be universal, not unique to PSiNPs.

Fig. R1. TEM images of Bel7402 cells after treatment with PSiNPs at a concentration of 200 µg/mL for different time intervals. Scale bar: 1 µm. Black arrows indicate autophagosome and red arrows indicate MVBs.

3. *In the western blot study with the exosomes purified by ultracentrifugation, the authors don't show any enrichment of CD63, which indicates that their purification process was not optimized. Further, they showed enrichment of Tsg101, which is not a membrane-bound protein, in their coated nanoparticle formulation. How do they explain these results?*

Response:

We thank the reviewer's comments. In the original manuscript, the exosomes we used were stored at 4 °C for several weeks, which might result in the deactivation of exosomes. In the revised manuscript, we modified the protocol of exosomes purification. The collected cell supernatants were stored at -80 °C immediately and re-dissolved at 4 °C overnight before ultracentrifugation for exosome purification. Western blot results showed that the enrichment of CD63 and Tsg101 was observed in exosomes and E-PSiNPs. Meanwhile, no obvious expression of calnexin, located in endoplasmic reticulum (ER), was detected in exosomes and E-PSiNPs, revealing that exosomes are coated on PSiNPs in E-PSiNPs.

In our work, PSiNPs were colocalized with FTIC-CD63-labeled MVBs after internalization and then exocytosed, indicating the similar exocytosis pathway with the natural exosomes. Therefore, E-PSiNPs not only enrich the membrane-bound protein, like CD63, but also enrich other exosome markers, like Tsg101.

According to the reviewer's suggestion, we added these new data in the revised manuscript, Fig. 2E and Supplementary Fig. 6.

4. For the nanoparticle characterization, why was the number distribution used instead of intensity distribution. The Z-average sizes of both the uncoated and coated particles, along with the PDI of each, should be reported. How can the authors explain that the size distribution of the coated particles is better than the uncoated particles? How do the authors explain the large size increase on DLS, whereas their TEM image shows minimal size increase?

Response:

We thank the reviewer's constructive suggestion. In the revised manuscript, we used the intensity distribution of PSiNPs and E-PSiNPs instead of number distribution. Meanwhile, we provided the Z-average sizes of PSiNPs and E-PSiNPs and their corresponding PDI from DLS analyses. The average size of PSiNPs and E-PSiNPs was 150 ± 11 nm and 260 ± 15 nm, and PDI was 0.208 ± 0.028 and 0.145 ± 0.032 , respectively. We added the new data in the revised manuscript, Page 9 and Fig. 2A.

In our manuscript, we used the repeated centrifugation to collect E-PSiNPs, which might make the size distribution of E-PSiNPs better than PSiNPs.

In the case of size measurement by TEM and DLS analysis, TEM images depict the size at the dried state of samples, while the size measured by DLS is a hydrodynamic diameter (hydrated state), and therefore the samples show a larger hydrodynamic volume due to solvent effect in the hydrated state. In the original manuscript, the size of PSiNPs and E-PSiNPs was a little different using DLS and TEM analysis, which might be due to the process involved in the preparation of samples for measurement.

5. The fluorescence of DiO/FITC and DOX are not well separated. It is surprising that the authors were able to resolve these fluorophores. Are the authors certain that their imaging data using these dyes are reliable?

Response:

We thank the reviewer's critical comments. The optimal excitation wavelength of DiO/FITC and DOX was 488 nm and 559 nm, respectively, as shown in their absorbance spectra (**Fig. R2A**). We measured the emission spectra of DiO/FITC and DOX between 500-520 nm at the excitation wavelength of 488 nm. In addition, we also measured the emission spectra of DOX and DiO/FITC between 570-600 nm at the excitation wavelength of 559 nm. In the DOX detection range (red region), the fluorescence of DiO/FITC was negligible compared with DOX at the excitation wavelength of 559 nm, and in the

DiO/FITC detection range (green region), the fluorescence of DOX was negligible compared with DiO/FITC at the excitation wavelength of 488 nm, suggesting that the fluorescence of DiO/FITC and DOX can be well separated under these conditions. Therefore, in our original manuscript, we observed DiO/FITC at the excitation wavelength of 488 nm and the emission range of 500-520 nm, and DOX at the excitation wavelength of 559 nm and the emission range of 570-600 nm by confocal microscope. We supplemented the details on the confocal microscopic experiments in the Experimental Section in the revised manuscript.

Fig. R2. (A) The excitation wavelength of DOX and FITC/DiO. (B) Emission spectra of DOX and DiO/FITC at the excitation wavelength of 559 nm. (C) Emission spectra of DiO/FITC and DOX at the excitation wavelength of 488 nm.

6. How do the authors explain the poor drug loading efficiency? Loading at <1% efficiency is not economical for future translation. How did the authors ensure that their uncoated and coated formulations had equivalent drug loading before testing them *in vitro* and *in vivo*? It seems unlikely that this process can be controlled well enough to achieve equivalent loading given the complexity of the coating process. How did the authors ensure that they released all of the DOX from their nanoparticles when applying their HPLC protocol?

Response:

We appreciate the reviewer's critical comments. Indeed the drug loading efficiency of E-PSiNPs was 0.8%. However, the drug loading degree of DOX@E-PSiNPs was 300 ng DOX/ μ g protein (exosomes were quantified according to the protein content), which was about 2-3-fold higher than that of the recently reported drug-loaded exosomes (*ACS Nano* 2013, 7: 7698; *ACS Nano* 2016,10: 3323). In addition, in our system, the yield of exosomes increased by nearly 34-fold when the cells were treated with PSiNPs, and the anticancer activity of DOX@E-PSiNPs at DOX dosage of 0.5 mg/kg was stronger than free DOX at 4 mg/kg dosage (a common DOX use dosage). In view of the relatively high yield of E-PSiNPs and low DOX dosage used, E-PSiNPs as a drug carrier was promising in the future translation.

The drug loading efficiency of PSiNPs can be controlled by adding different concentrations of DOX into PSiNPs (**Fig. R3**). In our manuscript, when we performed the *in vitro* and *in vivo* experiments, the same amounts of DOX and PSiNPs in DOX@PSiNPs and DOX@E-PSiNPs were used by adjusting the same drug loading efficiency of PSiNPs as E-PSiNPs. For example, as we determined that the drug loading efficiency of E-PSiNPs was 0.8%, we prepared DOX@PSiNPs with the drug loading efficiency of 0.8% by loading 0.11 mg DOX into 10 mg PSiNPs.

Since PSiNPs were extremely unstable in base solution, DOX@PSiNPs and DOX@E-PSiNPs were quickly dissolved to release DOX in base solution. In our manuscript, when measuring DOX content in DOX@PSiNPs and DOX@E-PSiNPs by HPLC, DOX@PSiNPs and DOX@E-PSiNPs were first lysed in 1 M of NaOH for 30 min, followed by neutralization by 1 M HCl at the same volume. No obvious size of DOX@PSiNPs and DOX@E-PSiNPs was detected after treatment by DLS analysis, suggesting that DOX@PSiNPs and DOX@E-PSiNPs were completely dissolved and released all DOX. The similar method of measuring DOX content in PSiNPs was reported elsewhere (*Adv. Funct. Mater.* 2012, 22: 4225; *Adv. Mater.* 2014, 26: 7643).

Fig. R3. Drug loading efficiency of PSiNPs at the different DOX amounts added. Data were represented as mean \pm SD (n=3)

7. Their formulation is fairly large (almost 300 nm). It seems unlikely that such a large nanoformulation can penetrate tumors by traveling through the interstitial space. Is there another mechanism to explain their deep penetration?

Response:

We thank the reviewer's critical comments. As reported in our previous work (*ACS Appl. Mater. Inter.* 2016, 8: 27611), PSiNPs could be exocytosed from tumor cells and re-internalized by neighboring tumor cells, resulting in domino-like intercellular delivery of PSiNPs and deep tumor penetration. In the revised manuscript, we found that DOX@E-PSiNPs exhibited stronger intercellular delivery ability compared with DOX@PSiNPs, which might explain deeper tumor penetration of DOX@E-PSiNPs. According to the reviewer's suggestion, we added these new data and opinions in the revised manuscript, Page 18 and Supplementary Fig. 21.

8. The description of cancer stem cells as "side population cells" is confusing. Is this a common name for them?

Response:

We thank the reviewer's comments. Side population cells were reported to show characteristics of cancer stem cells (CSCs) responsible for drug resistance, metastasis and high tumorigenicity (*Adv. Drug. Deliver. Rev.* 2013, 65: 1763; *Proc. Natl. Acad. Sci USA* 2002, 99: 12339). In the original manuscript, to determine the accumulation of DOX@E-PSiNPs in CSCs and their CSCs killing activity *in vivo*, we measured the DOX content in side

population cells of tumors and the proportion of side population in H22 tumor-bearing mice. In the revised manuscript, to evaluate the inhibitory effects of DOX@E-PSiNPs on cancer stem cells *in vivo*, we not only determined the number of side population cells, but also the number of CD133⁺ cells in H22 tumor-bearing mice after intravenous injection of PBS, E-PSiNPs, free DOX, DOX@PSiNPs or DOX@E-PSiNPs at 0.5 mg/kg DOX dosage, or high dosage of DOX at 4 mg/kg once every three days for 17 days, since CD133 was reported to be a cancer stem cell marker for hepatocellular carcinoma (*Nat. Med.* 2011, 17: 313; *Cell Res.* 2012, 22: 259). The results showed that the number of both side population cells and CD133⁺ cells was the lowest in DOX@E-PSiNPs-treated group compared with the other groups. Furthermore, the tumor cells (10⁶ cells/mouse) of H22 tumor-bearing mice after treatment were subcutaneously transplanted into BALB/c mice. Only 33% of mice generated tumors at 40 days after secondary transplantation of tumor cells of DOX@E-PSiNPs-treated group, whereas 100% of mice generated tumors at 6 days after secondary transplantation of tumor cells of PBS-, E-PSiNPs-, free DOX- or DOX@PSiNPs-treated group. These results strongly suggest that DOX@E-PSiNPs can significantly decrease the number of CSCs.

According to the reviewer's suggestion, we added the new data in Page 22 and Figures 6D,E,H.

9. *The authors may be overstating their claim that traditional methods for coating membrane onto nanoparticles are disruptive.*

Response:

We thank the reviewer's pertinent comments. In the revised manuscript, we have modified the description of traditional methods for coating membrane onto nanoparticles. Please see Page 3 in the revised manuscript.

10. *Can the authors provide the absorbance profile and the emission profile at the appropriate excitation wavelength of their porous silicon nanoparticles?*

Response:

According to the reviewer's suggestion, we added the absorbance profile of PSiNPs and their emission profile at the excitation wavelength of 488 nm in the revised manuscript, Page 6 and Supplementary Fig. 1D.

11. *Can the authors demonstrate the colocalization of their nanoparticles with LC3 in Figure 1B?*

Response:

We appreciated the reviewer's constructive suggestion. In the revised manuscript, we added the co-localization of E-PSiNPs with LC3 in Fig. 1B.

12. *For their survival curves, what was the exact statistical test that was applied? If by the log-rank test, it seems highly unlikely that the difference between their last two groups was significant for either experiment. Can they provide the actual p-values?*

Response:

We re-analyzed the statistical significance of the survival curves of different groups by log-rank test and found that there was not statistically significant between DOX@E-PSiNPs- and high dosage of free DOX-treated group. However, compared with high dosage of free DOX-treated group, the survival time of DOX@E-PSiNPs-treated group improved from 109 day to 122 day in H22 tumor-bearing mice, and from 39 day to 43 day in B16-F10 tumor-bearing mice, respectively. We revised the data and supplemented the related information in the revised manuscript, Page 21 and Fig. 6C, 7C.

RESPONSE TO REVIEWER #2:

This is a well-done, multi-faceted study involving an innovative approach to targeted anticancer therapy using an exosome sheath to provide recognition for a DOX-loaded porous silicon drug delivery platform. The authors describe outcomes from a rational series of experiments – demonstration of efficient cellular uptake by cancer cells/CSCs, along with strong cytotoxicity and enhanced tumor accumulation and penetration-whose conclusions build iteratively in a logical manner. This is a significant advance in retention of surface protein selectivity and (hopefully) stability in nanoscale drug delivery in useful anti-cancer therapies.

I very much enjoyed reading and analyzing this manuscript. I do have some focused questions that I would like the authors to address before publication can be recommended:

1. The porous Si platform is a key component in controlled, sustained delivery. A key question that requires additional commentary in this manuscript is both the morphology/microstructure of the initial porous Si particles (TEM images + size distribution data) along with evolution of the porous Si morphology, if any, before/after incorporation into the exosome construct? [hydrodynamic radii data are provided, but is there any change in morphology of individual porous Si particles (on a statistical basis, observed via SEM/TEM?)]. OES ICP measured total dissolved Si content, but does not provide useful structural data. A second related question concerns any evidence (or lack thereof) in porous Si particle morphology during the measured therapeutic windows utilized in these experiments?

Response:

We appreciated the reviewer's constructive comments. In the revised manuscript, TEM was used to observe the morphology of PSiNPs and E-PSiNPs on a statistical basis. The results showed that the obvious membrane was found to coat onto PSiNPs in E-PSiNPs. Furthermore, we observed that after incubation of DOX@E-PSiNPs in PBS for 72 h, the morphology of DOX@E-PSiNPs did not change. In addition, no significant degradation of DOX@E-PSiNPs in PBS was detected. We added these new data in the revised manuscript, Page 9, Page 12 and Figs. 2B, 3D, and Supplementary Fig. 9.

2. Porous Si nanoparticle degradation in vitro/in vivo is quite sensitive to feature size and surface chemistry. It can be quite fast (hours) or slow (weeks to months), depending on the above factors. Neither of these key pieces of information are readily noted in the manuscript in its present form, only that the particles are unchanged after 6 days exposure in vitro (page 12). This sort of detail is important to understanding the long term behavior of this material in vivo.

Response:

We thank the reviewer's constructive comments. In the revised manuscript, we supplemented the characterization of PSiNPs, such as size, morphology, BET surface area, pore volume and average pore diameter. Meanwhile, we determined the degradation behavior of PSiNPs, E-PSiNPs and DOX@E-PSiNPs in PBS and the morphology change of DOX@E-PSiNPs after incubation in PBS. Our results showed that very little degradation and no significant morphology change of DOX@E-PSiNPs was detected in PBS buffer, suggesting that DOX@E-PSiNPs were relatively stable.

According to the reviewer's suggestion, we added the new data in the revised manuscript, Pages 6, Pages 12, Fig. 3D, and Supplementary Fig. 1, 9.

3. Other than sustained delivery, is there evidence for any unique structural contribution of the nanoscale Si carrier? If so, these points should be emphasized here.

Response:

We thank the reviewer's pertinent comments. PSiNPs induce autophagy in cancer cells. Upon autophagy induction, cytoplasmic material is sequestered in double-membrane vesicles termed autophagosomes, which can fuse with MVBs to form amphisomes or directly deliver to the lysosomes for degradation. Thus, the induction of autophagy usually inhibits the release of exosomes (*Cell. Mol. Life Sci.* 2018, 75, 193). However, when the cells do not degrade material in the lysosomes due to the lysosomal defect, lysosomal overload or transport interference, the contents of lysosomes, MVBs or amphisomes are exocytosed as exosomes when fusing with cell membrane (*Cell. Mol. Life Sci.* 2018, 75, 193). According to the previous report (*Adv. Colloid. Interface Sci.* 2012, 175:25) and our data, we found that PSiNPs were stable and did not degrade at lysosomal acidic pH. This unique structure promotes cancer cells to release exosome-coated PSiNPs when PSiNPs were incubated with cancer cells. We

added the new data and the opinions in the revised manuscript, Page 29 and Supplementary Fig. 29.

4. In this regard, the authors only address three references to porous Si drug delivery platforms (unless I am mistaken), and there needs to be more added. There is no shortage of possible choices, as the authors are likely aware.

Response:

We thank the reviewer's constructive suggestion. In the revised manuscript, we added more references about porous Si drug delivery platforms as reference 25-31.

5. Ultimate clinical relevance is dependent on multiple factors, of course, one of which is shelf-life. What is the stability of a exosome- packaged, DOX-loaded porous Si material?

Response:

We thank the reviewer's constructive suggestion. In the revised manuscript, we evaluated the change of size, zeta-potential and cell internalization ability of DOX@E-PSiNPs after storage at $-80\text{ }^{\circ}\text{C}$ for 1 month or lyophilization followed by resuspension in PBS 1 week later. The results revealed that no significant change was detected in the size, zeta-potential and cellular uptake of DOX@E-PSiNPs by cancer cells under these two storage conditions, suggesting that DOX@E-PSiNPs were stable under these two storage conditions, which benefits further translation and clinic applications.

According to the reviewer's suggestion, we added these new data in the revised manuscript, Pages 12 and 14, and Supplementary Fig. 8.

RESPONSE TO REVIEWER #3:

This work represents physical and chemical characterization of silicon-based, exosome-encapsulated nanoparticles loaded with doxorubicin, and in vitro and preclinical in vivo proof of concept studies using hepatocellular carcinoma. The manuscript is reasonably well written but there are several errors in English grammar and spelling which should be corrected. Furthermore, from a model and therapeutic point-of-view, it would be useful for the authors to address the following:

1. The Dox@E-PSiNPs are generated from a human hepatocellular carcinoma line, but are delivered to murine species. What is the immunocompetence of the murine models and have the authors performed any immunologic assessments to determine if a cellular or humoral immune response develops? How would the authors propose avoiding an immunologic reaction in human patients treated with these exosome-encapsulated NPs?

Response:

We thank the reviewer's critical question. In our animal experiment, we used DOX@E-PSiNPs exocytosed from H22 cells, which originated from ascitic fluid of BALB/c mice, to treat B16-F10 tumor-bearing C57BL/6 mice. In the revised manuscript, we analyzed the immune response in C57BL/6 mice after intravenous injection of PBS, E-PSiNPs, free DOX, DOX@PSiNPs or DOX@E-PSiNPs exocytosed from H22 cells at 0.5 mg/kg DOX dosage, or high dosage of free DOX at 4 mg/kg. IgM is the marker of innate immunity (*Cell Mol. Immunol.* 2013, 10: 113). TNF- α , IL-1 β and IL-6 play important roles in the regulation of immune response (*Nat. Immunol.* 2015, 16:448; *Nat. Rev. Drug Discov.* 2003, 2: 736; *Nat. Immunol.* 2015, 16: 343). The results showed that no significant change in the contents of IgM, TNF- α , IL-1 β and IL-6 was detected in serum of DOX@E-PSiNPs-treated mice compared with the other groups, suggesting that DOX@E-PSiNPs exocytosed from H22 cells did not induce immune response in C57BL/6 mice. We added these new data in the revised manuscript, Page 25 and Supplementary Fig. 28.

In our manuscript, we found that DOX@E-PSiNPs, regardless of their origin, exhibited cross-reactive cellular uptake and cytotoxicity against bulk tumor cells and CSCs. However, if DOX@E-PSiNPs are used to treat tumor patients in the future, we can use tumor patient-derived DOX@E-PSiNPs for personalized treatment to avoid all possible risks.

2. Figure 5 indicates that a high level of Dox@E-PSi-NPs is discovered in non-malignant liver, actually at much higher levels than are seen in the malignant hepatic tumors. What cells in the liver are accumulating these NPs; Kupffer cells, hepatocytes, or some other cell type? Is there any liver toxicity, inflammation, or sub-acute liver injury noted in the animals treated repeatedly with these NPs?

Response:

We understand the reviewer's concern on the side effects of DOX@E-PSiNPs. In our manuscript, DOX@E-PSiNPs could significantly inhibit tumor growth compared with free DOX or DOX@PSiNPs at DOX dosage of 0.5 mg/kg, even stronger than free DOX at 4 mg/kg dosage. In the revised manuscript, we determined the DOX bio-distribution in H22 tumor-bearing mice after intravenous injection of free DOX, DOX@PSiNPs or DOX@E-PSiNPs at DOX dosage of 0.5 mg/kg, or high dosage of DOX at 4 mg/kg. The results showed that although DOX@E-PSiNPs were accumulated in the liver, spleen and kidney at relatively high levels, their accumulation in these normal tissues were significantly lower than high dosage of free DOX at 4 mg/kg. Importantly, DOX@E-PSiNPs exhibited stronger tumor targeting capacity compared with free DOX, DOX@PSiNPs at 0.5 mg/kg DOX dosage, comparable to high dosage of free DOX at 4 mg/kg.

It has been reported that nanoparticles were easily phagocytosed by Kupffer cells after intravenous injection (*Proc. Natl. Acad. Sci. USA.* 2017, 114: E10871; *J. Control. Release* 2012, 161: 164; *Nano Today* 2015, 10: 11). Similarly, we found that DOX@E-PSiNPs were mostly accumulated in CD68-stained Kupffer cells in liver at 24 h after intravenous injection. However, no obvious toxicity in major organs (including heart, liver, spleen, lung and kidney), as evidenced by hematoxylin-eosin (H&E) staining and serological analysis, was detected in DOX@E-PSiNPs-treated group, although high dosage of DOX at 4 mg/kg exhibited significant heart toxicity.

According to the reviewer's suggestion, we added these new data in the revised manuscript, Pages 17 and 21, and Fig. 5A, Supplementary Figs. 18, 25, 26.

3. *Treatment of hepatocellular carcinoma or melanoma with doxorubicin has limited success clinically, primarily due to induction of P-glycoprotein with increased levels of this trans-membrane protein responsible for chemotherapy resistance following doxorubicin exposure. Have the authors evaluated P-glycoprotein expression in their cancer cell lines, including CSCs, following treatment with the doxorubicin-loaded NPs? The authors results demonstrate improved response and prologed survival in vitro and in animals treated with the novel NPs, but no complete response is seen. Can the authors offer any data explaining the lack of any complete responders? Is this due to CSCs developing a resistant phenotype?*

Response:

We thank the reviewer's constructive suggestion. In the revised manuscript, we determined P-glycoprotein (P-gp) expression in H22 CSCs after treatment with PBS, free DOX, DOX@PSiNPs or DOX@E-PSiNPs. The results showed that DOX@E-PSiNPs treatment decreased P-gp expression in H22 CSCs compared with PBS, free DOX and DOX@PSiNPs. Meanwhile, we determined DOX fluorescence in H22 CSCs after treatment with free DOX, DOX@PSiNPs or DOX@E-PSiNPs for 2 h, followed by washing with PBS and then incubating in fresh media for different time courses. More DOX was retained in DOX@E-PSiNPs-treated CSCs compared with that in free DOX- or DOX@PSiNPs-treated group, further confirming that DOX@E-PSiNPs might increase intracellular DOX accumulation by decreasing P-gp expression to inhibit DOX efflux. We added the new data in the revised manuscript, Page 14 and Supplementary Fig. 10.

In our manuscript, we found that DOX@E-PSiNPs significantly inhibited tumor growth and extended survival time of tumor-bearing mice. However, DOX@E-PSiNPs treatment did not induce complete response in tumor-bearing mice, which might be probably due to that DOX@E-PSiNPs did not completely eradicate CSCs in tumor tissues. To achieve the complete response, we can combine DOX@E-PSiNPs treatment with immunotherapy, such as anti-PD-1 antibody or anti-CTLA-4 antibody in future research. We added these opinions in the Discussion section in the revised manuscript, Page 28.

RESPONSE TO REVIEWER #4:

In this study, Yong et al. developed a novel biocompatible exosome sheathed PSiNPs (ePSiNPs) as a delivery vehicle for targeted cancer chemotherapy. The authors demonstrated autophagy is involved in exocytosis of PSiNPs. Furthermore, they showed that the exocytosed PSiNPs is sheathed by exosomes. Once the PSiNPs was loaded with doxorubicin and intracellularly processed to form DOX@ePSiNPs, it shows stronger uptake efficiency and cytotoxicity for cancer stem cells (CSCs). In vivo study reveals that DOX@E-PSiNPs exhibits the highest anti-cancer effect in comparison to free doxorubicin and DOX@PSiNPs.

1. Is the side population confirmed to be CSCs in these tumor cell lines? Authors should also test other verified CSC markers for these tumor models.

Response:

We thank the reviewer's critical question. Side population cells were reported to show characteristics of cancer stem cells responsible for drug resistance, metastasis and high tumorigenicity (*Adv. Drug. Deliver. Rev.* 2013, 65: 1763; *Proc. Natl. Acad. Sci. USA.* 2002, 99: 12339). In the revised manuscript, to evaluate the inhibitory effects of DOX@E-PSiNPs on cancer stem cells, we determined not only the number of side population cells, but also the number of CD133⁺ cells in H22 tumor-bearing mice after treatment with PBS, E-PSiNPs, free DOX, DOX@PSiNPs or DOX@E-PSiNPs at 0.5 mg/kg DOX dosage, or high dosage of DOX at 4 mg/kg once every three days for 17 days, since CD133 was reported to be a cancer stem cell marker for hepatocellular carcinoma (*Nat. Med.* 2011, 17: 313; *Cell Res.* 2012, 22: 259). The results showed that the number of side population cells and CD133⁺ cells was the lowest in DOX@E-PSiNPs-treated group compared with the other groups.

According to the reviewer's suggestion, we added the new data in the revised manuscript, Page 22 and Fig. 6D.

2. Authors showed that DOX@E-PSiNPs can shrink the tumor very fast and also decreased the CSC percentage significantly more than other three controls. But, the CSC percentage change doesn't mean the absolute CSC number change which is based on the total tumor cell number in the tumor. The golden standard to measure the CSC change is to do the secondary transplantation from the primary tumors. It looks like DOX@E-PSiNPs target both CSC and non-CSC simultaneously, if the absolute CSC number is higher

in DOX@E-PSiNPs treated group, it might promote the tumor relapse and metastasis.

Response:

We thank the reviewer's critical questions. In the revised manuscript, we constructed GFP-expressing H22 cells and subcutaneously injected them to the flanks of BALB/c mice. When tumors grew around 200 mm³, the mice were intravenously injected with PBS, E-PSiNPs, free DOX, DOX@PSiNPs or DOX@E-PSiNPs at 0.5 mg/kg DOX dosage, or high dosage of free DOX at 4 mg/kg once every three days for 17 days. The number of SP cells in GFP-positive H22 cells was determined by flow cytometry. The results showed that the number of side population cells in DOX@E-PSiNPs-treated group was significantly lower than the other groups including high dosage of free DOX-treated group. Here, we used the unit of side population cell number as side population cell number per gram tumor tissue. Due to the significantly decreased tumor weight in DOX@E-PSiNPs-treated group compared with other groups, the total side population cells in DOX@E-PSiNPs should be much lower than other groups.

In addition to further confirm that DOX@E-PSiNPs efficiently kill of CSCs *in vivo*, tumor tissues of H22 tumor-bearing mice after treatment were dispersed into single cell suspension, and the same amounts of cells (10⁶ cells/mouse) were subcutaneously transplanted into BALB/c mice, according to the reviewer's suggestion. 100% of mice (6/6 mice) generated tumors at 6 days after secondary transplantation of tumor cells from PBS-, E-PSiNPs-, free DOX- or DOX@PSiNPs-treated group. However, 83.3% (5/6 mice) and 33.3% (2/6 mice) of mice generated tumors at 40 days after secondary transplantation of tumor cells from high dosage (4 mg/kg) of free DOX- and DOX@E-PSiNPs-treated groups (0.5 mg/kg), respectively. These results strongly showed that DOX@E-PSiNPs significantly reduced the number of CSCs.

According to the reviewer's suggestion, we added the new data in the revised manuscript, Page 22 Line 11-14, Line 17-22 and Fig. 6E, 6H.

3. "CSCs located within the hypoxic core of tumor mass" on Page 15 is not exactly right since it's been confirmed that some metastatic CSCs are located in the edge of the tumors.

Response:

We thank the reviewer's comment. According to the reviewer's suggestion, we have deleted the sentence "CSCs located within the hypoxic core of tumor mass" in the revised manuscript.

4. The claim "Significantly fewer metastatic nodules were detected in the DOX@E-PSiNPs-treated group" has to be further demonstrated that fewer metastatic nodules was due to DOX@E-PSiNPs targeting the metastatic tumor cells from the primary cells or targeting the tumor cells in the lung?

Response:

We thank the reviewer's critical comments. In this manuscript, we constructed lung metastatic tumor model by intravenously injecting 5×10^5 B16-F10 cells into C57BL/6 mice. To evaluate why significantly fewer metastatic nodules were detected in the DOX@E-PSiNPs-treated group, in the revised manuscript we determined DOX content in lung metastatic nodules of mice intravenously injected with free DOX, DOX@PSiNPs or DOX@E-PSiNPs at 0.5 mg/kg DOX dosage, or free DOX at high dosage of 4 mg/kg. The results showed that more DOX was accumulated in lung metastatic nodules of DOX@E-PSiNPs-treated group than that of free DOX- or DOX@PSiNPs-treated group, comparable with that of high dosage of free DOX-treated group, suggesting that DOX@E-PSiNPs can effectively target the tumor cells in the lung.

According to the reviewer's suggestion, we added these new data in Page 17 and Supplementary Fig. 19.

5. The use of cell lines for the study is not well designed and confusing, for example, in Fig 1 through Fig. 3, they used Bel7402 to test the involvement of autophagy and so on, somehow they skipped to H22 and B16-F10 in Fig 4 to test cellular uptake and cytotoxicity of DOX@E-PSiNPs without explanation. Similarly, they applied H22 to in vivo model to test the enrichment and CSCs killing activity of DOX@ePSiNPs, then used B16-F10 for the lung metastasis model. It's understandable that authors want to test in different tumor types, but one tumor type should be tested for all studies in the paper and other tumor types can be tested for a support.

Response:

We thank the reviewer's critical comments. In this manuscript, we determined the autophagy-involved exocytosis of exosome-coated PSiNPs (E-PSiNPs) for efficient anticancer drug delivery. First, we determined the PSiNPs-induced autophagy was involved in the exocytosis of E-PSiNPs. Considering that adherent cells were easier to operate, we used Bel7402 cells to systematically determine the PSiNPs-induced autophagy and the exocytosis of E-PSiNPs by using western blot, confocal microscope and promoter/inhibitor treatment. Furthermore, we used H22 cells to confirm that PSiNPs-induced autophagy and the subsequent exocytosis of E-PSiNPs were universal by western blot (Supplementary Figs. 3 and 6 in the revised manuscript). In addition, we confirmed the exocytosis of DOX@E-PSiNPs in Bel7402 and H22 cells (Fig. 3A and Supplementary Fig. 7 in the revised manuscript).

Then, we determined the biological effects of DOX@E-PSiNPs. We used H22, Bel7402 and B16-F10 cells to evaluate the cross-reactive cellular uptake and cytotoxicity against bulk cancer cells and CSCs (Fig. 4 and Supplementary Figs. 11-15 in the revised manuscript). Furthermore, we used H22 tumor-bearing mice to determine the tumor accumulation, tumor penetration and intracellular accumulation of DOX@E-PSiNPs in total tumor cells and side population with the properties of CSCs (Fig. 5 in the revised manuscript), as well as their anticancer efficacy and CSCs killing activity (Fig. 6 in the revised manuscript). Furthermore, to confirm the cross-reactive anticancer and CSCs killing efficacy of DOX@E-PSiNPs, we used DOX@E-PSiNPs exocytosed from H22 cells to treat B16-F10 tumor-bearing mice, and found that DOX@E-PSiNPs exocytosed from H22 cells effectively inhibited the lung metastasis and had CSCs killing activity (Fig. 7 in the revised manuscript).

Therefore, H22 cells were used for all studies in this manuscript. In addition, we used different cancer cells, including B16-F10 and Bel7402 cells to confirm the notion that autophagy-involved exocytosis of DOX@E-PSiNPs for efficient cross-reactive anticancer and CSCs killing efficacy.

6. For Fig 2C and 2D, Dio and anti-CD63 antibody should also be incubated with non-exocytosed PSiNPs as negative control to exclude the possibility of non-specific binding.

Response:

We thank the reviewer's constructive suggestion. According to the reviewer's suggestion, we incubated non-exocytosed PSiNPs with DIO or anti-CD63 antibody as the negative controls. We added these new data in the revised manuscript, Figs. 2C and 2D.

7. For Fig 2E, is β -actin a proper internal control for both of total cell lysates and exosomes? The authors can display a ponceau S stained membrane as the reference of loaded protein.

Response:

We thank the reviewer's constructive suggestion. According to the reviewer's suggestion, we used Coomassie Blue to stain SDS-PAGE gels to visualize the protein bands. We added these data in the revised manuscript, Fig. 2E and Supplementary Fig. 6.

8. In Fig 5A, DOX@E-PsiNPs shows very high accumulation of DOX in liver, kidney and spleen (even higher than free DOX). This result indicates potential side effects of ePSiNPs as delivery vehicle for chemotherapy. The authors should discuss how to improve the specificity of ePSiNPs for targeting tumor cells.

Response:

We thank the reviewer's critical comments. In the revised manuscript, we treated H22 tumor-bearing mice with free DOX, DOX@PSiNPs or DOX@E-PSiNPs at DOX dosage of 0.5 mg/kg, or high dosage of DOX at 4 mg/kg. The results showed that although more DOX was located in liver, spleen and kidney of DOX@E-PSiNPs-treated mice compared with that in tumor, DOX accumulation in heart, liver, lung and kidney of DOX@E-PSiNPs-treated group was lower than that of high dosage of DOX-treated group (the anticancer activity of DOX@E-PSiNPs at 0.5 mg/kg was better than that of free DOX at 4 mg/kg). Furthermore, body weight, hematoxylin-eosin (H&E) staining of major organs (heart, liver, spleen, lung and kidney) and serological analysis showed that DOX@E-PSiNPs did not

cause obvious toxicity. We added these new data in Pages 17 and 21, Fig. 5A, and Supplementary Figs. 18, 24, 25, 26.

We agreed with the reviewer that it was better to improve the specificity of E-PSiNPs for targeting tumor cells. To achieve this, we can modify different targeting molecules, such as folic acid, RGD, et al, on exosome membrane in a future research work.

9. For Fig 5C, the authors claimed that DOX delivered with DOX@E-PsiNPs is detectable at deeper location, up to 400 μm from blood vessel in the text, somehow in the figure, the signal of DOX@E-PsiNPs is only detectable at the depth of 100 μm . In the contrast, DOX@PsiNPs is detectable at the depth of 400 μm .

Response:

In the original manuscript, we reversed the color of DOX@PSiNPs and DOX@E-PSiNPs by mistake. We corrected this mistake in the revised manuscript, as shown in Fig. 5C.

10. For Fig 6D, according to the description of procedure, cancer cells were not isolated from stroma cells. Thus the SP cells may not represent CSCs population. To address this question, the authors can label H22 cells with GFP for injection. GFP positive cells can be gated prior to the analysis of SP cells.

Response:

We thank the reviewer's constructive suggestion. In the revised manuscript, we constructed H22 cells stably expressing GFP. GFP-expressing H22 cells (10^6 cells) were subcutaneously injected to the flanks of BALB/c mice. When tumor volume reached around 200 mm^3 , the mice were administrated with PBS, E-PSiNPs, free DOX, DOX@PSiNPs or DOX@E-PSiNPs at DOX dosage of 0.5 mg/kg, or high dosage of DOX at 4 mg/kg once every three days for 17 days. To analyze the number of side population cells after treatment, the mice were sacrificed and the tumor tissues were digested into single cells. GFP-positive H22 cells were gated and the number of side population cells in GFP-positive H22 cells were measured by flow cytometry (450/45 BP filter for blue fluorescence and 660/20 BP 776 filter for red fluorescence). The results showed that DOX@E-PSiNPs significantly decreased the number of side population cells compared with free DOX, DOX@PSiNPs, or even high dosage of DOX.

According to the reviewer's suggestion, we added these new data in the revised manuscript, Page 22 Line 11-14 and Fig. 6E.

11. *Normal cancer cells should be changed to bulk cancer cells.*

Response:

We thank the reviewer's constructive suggestion. We changed "*normal cancer cells*" to "*bulk cancer cells*" in the revised manuscript.

12. *"Discussion" section is too simple or missing.*

Response:

We added a new discussion section in the revised manuscript.

13. *For Fig 1B, actual length of the scale bar was not described in figure legends.*

Response:

We thank the reviewer's constructive suggestion. According to the reviewer's suggestion, we added the description of scale bar in Fig. 1B in the revised manuscript.

14. *For Fig 3D, statistical significance should be displayed at the last time point.*

Response:

We thank the reviewer's constructive suggestion. According to the reviewer's suggestion, we added the statistical significance at the last time point in Fig. 3E in the revised manuscript (Fig. 3D in the original manuscript).

15. *For Fig 5A, labels of X-axis is not well displayed (partially hidden).*

Response:

We improved the images of Fig. 5A in the revised manuscript.

16. For Fig 5B, the meaning of the white bars on pictures of DOX group was not described in figure legends.

Response:

We added the description of white bars in the figure legend of Fig. 5B in the revised manuscript.

17. In Supplementary Fig. 8, the authors didn't show the data regarding the uptake efficiency of DOX@E-PsiNPs by B16-F10, just like the results in supplementary Fig. 7 for H22 and Bel7402. Does this mean DOX@E-PsiNPs exhibit no higher intracellular internalization than free DOX and DOX@PSiNPs in B16-F10? This question should be addressed clearly.

Response:

We thank the reviewer's constructive comments. In the revised manuscript, we evaluated the internalization and cytotoxicity of DOX@E-PSiNPs exocytosed from H22 cells against B16-F10 cells. In line with the results in H22 and Bel7402 cells using DOX@E-PSiNPs exocytosed from H22 cells and Bel7402 cells, respectively, DOX@E-PSiNPs exocytosed from H22 cells exhibited the highest internalization into B16-F10 cells and had the corresponding strongest cytotoxicity against B16-F10 cells. Similar results were obtained in H22 cells treated with DOX@E-PSiNPs exocytosed from B16-F10 cells. These results showed that DOX@E-PSiNPs exhibited cross-reactive cellular uptake and cytotoxicity against cancer cells. According to the reviewer's suggestion, we added these new results in the revised manuscript, Page 15 and Supplementary Fig. 15.

Reviewers' comments:

Reviewer #1 (Remarks to the Author):

Overall, the revised manuscript and supporting information are hard to follow without any track changes considering so many new data and figures are involved. The authors are strongly advised to highlight the changes in the manuscripts and connect the changes being made to the response letter.

In general, the authors have done a good job in addressing the prior comments by conducting new experiments. Most of the comments have been addressed. However, there are still a few that remain unclear.

1. In response to the mechanism of particle formation via exocytosis, the images are not convincing. TEM is not capable to reveal the formation process. While the fluorescent images show colocalization of the silica particles and phagosomes, it's shown that only few particles are internalized in the autophagosomes per cell. Will there be a significant issue in terms of production yield?
2. The argument of potential deactivation of exosomes causing weak CD63 intensity on western blot cannot be easily accepted. Unless the deactivation cause protein loss, it would not affect the western blot intensity of the protein.
3. Regarding drug loading, the authors argue that 300 ng DOX/ug protein in the formulation, but the reality is that protein counts only a small weight percentage in their formulation. Taking the silica particles and all other excipient materials (membrane lipids as well) into consideration, low drug loading yield is the issue. The authors need to consider/estimate the weight dosage of the nanodrug formulation in order to make the encapsulated drug substance reaching therapeutic concentration.

Reviewer #2 (Remarks to the Author):

The authors have carefully considered points of concern raised in my initial review of the original manuscript; the revised version with additional data is a healthy improvement.

One point remains, however. In the revised Figure 3 (specifically Fig 3d), the degradation profiles of the various platforms are shown. However, in Fig 3D the data points for DOX@E-PSINPs in PBS are not visible. Is this an oversight, or is the extent of degradation so small that it cannot be seen? Please clarify.

An interesting advance in targeted drug delivery, Publication is recommended if this remaining point is resolved.

Reviewer #3 (Remarks to the Author):

This is an interesting study using exosome-sheathed porous silicon nanoparticles that the authors chose to load with doxorubicin to test anti-cancer activity *in vitro* and *in vivo*. The paper is reasonably well written but should be edited for some minor errors in English grammar and word use; for example the authors use the words extravagation rather than extravasation. Exosomes are currently being studied both as biomarkers of disease states and incorporated into therapeutic delivery platforms. The paper has useful information and data but would be much stronger if the authors can address the following points:

1. The authors used an ectopic tumor model for their *in vivo* primary tumor studies. these tumors generally lack a realistic multi-cell-type microenvironment to model drug delivery accurately. Do the authors have data with orthotopic models in immune competent animals which includes the stromal elements present in human cancers?
2. The authors have provided little information on doxorubicin-release kinetics and possible cardio- and bone marrow toxicity. Doxorubicin is an interesting choice for cytotoxic agent to be studied; why did the authors select this drug and do they have toxicity data?
3. Exosomes bear the surface proteins derived from the membranes of origin. The authors provide no information on the proteins in their exosome platform. Have they performed antibody microarray analysis or other studies to characterize and quantify the proteins on the surface? This is critical when considering immunogenicity and circulating time.
4. The authors state the exosome-sheathed nano-delivery platform has improved tissue distribution and penetration due to EPR effect. The particle characterization, including size and surface charge, does not explain this apparent improved distribution. Have the authors considered mechanistic

studies to determine critical factors that may enhance delivery in valid, realistic tumor microenvironments. Distribution mathematical modeling may be very helpful.

Reviewer #4 (Remarks to the Author):

The authors have addressed all of the questions clearly in the revised manuscript.

Reviewer #4's feedback on rebuttal to ref#3's concerns from last round:

I have read the author's response to the concerns from reviewer #3, and I think the authors addressed most of the concerns, but it should be addressed why DOX@E-PSiNPs decreases P-gp expression in CSCs compared with PBS, free DOX and DOX@PSiNPs. In addition, the question below was not well answered with the data support: "Can the authors offer any data explaining the lack of any complete responders? Is this due to CSCs developing a resistant phenotype?"

Response to Reviewers' comments

We would like to express our sincere thanks to all four reviewers for their critical and constructive comments. We have performed substantial additional experiments to address their concerns. We respond point-by-point to each of their comments and criticisms. We feel that their comments have helped us to significantly improve and strengthen the manuscript, and clarified the issues raised by the reviewers.

We hope that the revision has addressed their major concerns.

Response to Reviewer #1:

Overall, the revised manuscript and supporting information are hard to follow without any track changes considering so many new data and figures are involved. The authors are strongly advised to highlight the changes in the manuscripts and connect the changes being made to the response letter.

In general, the authors have done a good job in addressing the prior comments by conducting new experiments. Most of the comments have been addressed. However, there are still a few that remain unclear.

1. In response to the mechanism of particle formation via exocytosis, the images are not convincing. TEM is not capable to reveal the formation process. While the fluorescent images show colocalization of the silica particles and phagosomes, it's shown that only few particles are internalized in the autophagosomes per cell. Will there be a significant issue in terms of production yield?

Response: We thank the reviewer's critical comments. In our manuscript, confocal fluorescence microscopic images showed that treatment with PSiNPs led to significantly enhanced puncta formation of LC3-labeled vacuoles in cancer cells and the intracellular PSiNPs were mostly captured in the LC3⁺ autophagosomes (Figure 1B in the revised manuscript), confirming that PSiNPs induced autophagosome formation. We agreed with the reviewer that not too many PSiNPs were internalized in autophagosomes of each cell, which might affect the production yield of E-PSiNPs. In the future work, we can modify different targeting molecules, such as folic acid, RGD, etc., on the surface of PSiNPs to enhance the cellular uptake by cancer cells to induce more autophagy, thus generating more E-PSiNPs.

2. The argument of potential deactivation of exosomes causing weak CD63 intensity on western blot cannot be easily accepted. Unless the deactivation cause protein loss, it would not affect the western blot intensity of the protein?

Response: We thank the reviewer's critical comments. In the original manuscript, the exosomes we used were stored at 4 °C for several weeks and western blot analysis showed no enrichment of CD63 in exosomes and E-PSiNPs. Storage conditions, such as temperature, might destabilize the surface characteristics, morphological features and protein content of exosomes, generating significant impact on physicochemical and functional

properties of exosomes (*J. Extracell. Vesicles* 2017, 6: 1359478; *Biotechnol. Bioprocess Eng.* 2016, 21: 299-304). Lee *et al* also reported that incubation of exosomes at 4 °C and room temperature resulted in major loss of CD63 (*Biotechnol. Bioprocess Eng.* 2016, 21: 299-304). In the previously revised manuscript, we modified the protocol of exosomes purification. The collected cell supernatants were stored at -80 °C immediately and re-dissolved at 4 °C overnight before ultracentrifugation for exosome purification. Western blot results showed that the enrichment of CD63 and Tsg101 was observed in exosomes and E-PSiNPs (Fig. 2E and Supplementary Fig. 6 in the revised manuscript).

3. Regarding drug loading, the authors argue that 300 ng DOX/ug protein in the formulation, but the reality is that protein counts only a small weight percentage in their formulation. Taking the silica particles and all other excipient materials (membrane lipids as well) into consideration, low drug loading yield is the issue. The authors need to consider/estimate the weight dosage of the nanodrug formulation in order to make the encapsulated drug substance reaching therapeutic concentration.

Response: We thank the reviewer's critical comments. In our manuscript, the drug loading degree of DOX@E-PSiNPs was 300 ng DOX/μg protein (exosomes were quantified according to the protein content) and the drug loading efficiency was 0.8% determined by high performance liquid chromatography (HPLC). Although the low drug loading efficiency, DOX@E-PSiNPs were used at the DOX dosage of 0.5 mg/kg in the *in vivo* anticancer experiments, with similar anticancer activity and less toxicity compared with free DOX at 4 mg/kg dosage. The weight dosage of DOX@E-PSiNPs is 1.25 mg/mouse (20 g weight) if we injected them at DOX dosage of 0.5 mg/kg. Compared with other group's report on DOX loaded PSiNPs (*Nat. Mater.* 2009, 8: 331-336; *Small* 2010, 6: 2546-2552), the weight of nanodrug formulation was comparable (as listed in the Table below). Moreover, the drug loading condition can be optimized by adding different concentrations of DOX@PSiNPs to cancer cells to obtain the higher drug loading efficiency of DOX@E-PSiNPs.

Sample name	DOX-LPSiNPs	DOX-loaded magnetic luminescent porous Si NPs	DOX@E-PSiNPs	
Reference	Nat. Mater. 2009, 8: 331-336	Small 2010, 6: 2546-2552	Our manuscript	
Commonly used DOX Dosage	4 mg/kg	4 mg/kg	0.5 mg/kg	
Drug loading efficiency	4.19%	9.8 %	0.8%	
Injected weight/mouse (20 g weight)	DOX	80 µg	80 µg	10 µg
	Protein	---	---	33 µg
	PSiNPs	1826 µg	736 µg	1207 µg
	Nanodrug formulation	1906 µg	816 µg	1250 µg

Response to Reviewer #2:

The authors have carefully considered points of concern raised in my initial review of the original manuscript; the revised version with additional data is a healthy improvement.

One point remains, however. In the revised Figure 3 (specifically Fig 3d), the degradation profiles of the various platforms are shown. However, in Fig 3D the data points for DOX@E-PSiNPs in PBS are not visible. Is this an oversight, or is the extent of degradation so small that it cannot be seen? Please clarify. An interesting advance in targeted drug delivery, Publication is recommended if this remaining point is resolved.

Response: We thank the reviewer's constructive comments. In the original Fig.3D, DOX@E-PSiNPs had similar degradation profile compared with E-PSiNPs. The blue line of DOX@E-PSiNPs was overlapped with the red line of E-PSiNPs. In the revised manuscript, we improved the figure to make them visible.

Response to Reviewer #3:

This is an interesting study using exosome-sheathed porous silicon nanoparticles that the authors chose to load with doxorubicin to test anti-cancer activity in vitro and in vivo. The paper is reasonably well written but should be edited for some minor errors in English grammar and word use; for example the authors use the words extravagation rather than extravasation. Exosomes are currently being studied both as biomarkers of disease states and incorporated into therapeutic delivery platforms. The paper has useful information and data but would be much stronger if the authors can address the following points:

Response: We thank the reviewer's constructive suggestion. We have corrected the errors in English grammar and word use carefully in the revised manuscript.

1. The authors used an ectopic tumor model for their in vivo primary tumor studies. these tumors generally lack a realistic multi-cell-type microenvironment to model drug delivery accurately. Do the authors have data with orthotopic models in immune competent animals which includes the stromal elements present in human cancers?

Response: We thank the reviewer's constructive suggestion. According to this, in the revised manuscript, we evaluated the anticancer activity of DOX@E-PSiNPs in orthotopic 4T1 breast cancer mode. 4T1 cells (2×10^5) suspended in 50 μ L PBS were injected to the right mammary fat pad of female BALB/c mice. When the tumor volume reached 50-70 mm³, the mice were intravenously injected with PBS, E-PSiNPs, free DOX, DOX@PSiNPs or DOX@E-PSiNPs at DOX dosage of 0.5 mg/kg, or high dosage of DOX at 4 mg/kg once every three days for five times. Consistently, our results showed that DOX@E-PSiNPs exhibited the excellent anticancer activity in orthotopic 4T1 tumor-bearing mice, as evidenced by smaller tumor volume and weight and longer survival time, similar to high dosage of DOX (Fig.7A-C in the revised manuscript).

In addition, the fewest colony number and smallest colony size were formed in DOX@E-PSiNPs-treated group when the tumor cells after treatment were seeded in soft 3D fibrin gels (90 Pa) for 5 days (Fig. 7D,E in the revised manuscript), which was developed to select CSCs (*Nat. Mater.* 2012, 11: 734–741; *Cell Res.* 2016, 26: 713–727), suggesting that DOX@E-PSiNPs have strong CSCs killing activity. DOX@E-PSiNPs did not cause toxicity to 4T1

tumor-bearing mice, as evidenced by routine blood test, serological analysis and H&E staining of major organs. Combined with the other tumor model in the original manuscript, our results strongly suggest that DOX@E-PSiNPs have excellent anticancer and CSCs killing activity with less toxicity.

According to the reviewer's suggestion, we added these data in the Figure 7 in the revised manuscript.

2. The authors have provided little information on doxorubicin-release kinetics and possible cardio- and bone marrow toxicity. Doxorubicin is an interesting choice for cytotoxic agent to be studied; why did the authors select this drug and do they have toxicity data?

Response: We thank the reviewer's critical comments. In the revised manuscript, DOX release profile of DOX@E-PSiNPs was determined in PBS with or without 10% FBS at different pH values (7.4, 6.8 or 4.5). The results showed that DOX@E-PSiNPs exhibited pH-dependent DOX release and more DOX was released from DOX@E-PSiNPs under lysosomal acidic pH (Supplementary Fig. 18 in the revised manuscript). Furthermore, we determined the intracellular trafficking of DOX@E-PSiNPs in B16-F10 cells (Supplementary Fig. 17 in the revised manuscript). We observed that when cancer cells were incubated with DOX@E-PSiNPs, CFSE-labeled exosomes and DOX entered cancer cells together and then colocalized with lysosomes. DOX was translocated to nuclei over time. Taken together with these data, we conclude that after uptake of DOX@E-PSiNPs by cancer cells, DOX@E-PSiNPs enter lysosomes, where DOX@E-PSiNPs released DOX at low lysosomal pH, subsequently drug molecules entering the nucleus via the nuclear pore to exert the cytotoxicity.

We understand the reviewer's concern on the toxicity of DOX@E-PSiNPs. In the original manuscript, we determined the toxicity of DOX@E-PSiNPs in H22 tumor-bearing mice. Our results showed that no obvious toxicity of DOX@E-PSiNPs, including cardiotoxicity, was observed in H22 tumor-bearing mice, as evidenced by body weight, hematoxylin-eosin (H&E) staining of major organs and serological analysis (Supplementary Fig. 29-31 in the revised manuscript).

In the revised manuscript, we further determined their biosafety in 4T1 orthotopic tumor model and H22 subcutaneous tumor model. 4T1 tumor-bearing mice were intravenously injected with free DOX, DOX@PSiNPs

or DOX@E-PSiNPs at DOX dosage of 0.5 mg/kg (the used therapeutic dosage for DOX@E-PSiNPs), or high dosage of DOX at 4 mg/kg for 5 times. Routine blood test, serological analysis and H&E staining of major organs showed that no toxicity was observed in DOX@E-PSiNPs-treated group (Supplementary Fig. 34-36 in the revised manuscript). However, high dosage of DOX exhibited significant heart toxicity (a significant increase in CK and LDH activity in serological analysis, as well as pronounced neutrophil gathering and myocardial necrosis in heart slices of H&E staining analysis) and bone marrow toxicity (a significant decrease in platelet and white blood cell count and hemoglobin content). Similar results were observed in H22 tumor-bearing mice after intravenous injection of DOX@E-PSiNPs at 0.5 mg/kg or even 0.8 mg/kg for 5 times (Supplementary Fig. 33 in the revised manuscript).

Taken together, our results showed that DOX@E-PSiNPs exhibit negligible *in vivo* toxicity and are relatively safe.

In this manuscript, we selected DOX as a model drug considering that DOX is widely used as an anticancer drug, and it has intrinsic fluorescence, which will help to evaluate their tumor accumulation and penetration and cellular uptake by cancer cells.

According to the reviewer's suggestion, we added these results in the revised manuscript, Supplementary Fig. 17, 18, 33-36.

3. Exosomes bear the surface proteins derived from the membranes of origin. The authors provide no information on the proteins in their exosome platform. Have they performed antibody microarray analysis or other studies to characterize and quantify the proteins on the surface? This is critical when considering immunogenicity and circulating time.

Response: We thank the reviewer's critical comments. Surface proteins of exosomes originated from cancer cells had been widely studied (*Nat. Cell Biol.* 2015, 17: 816-826; *Nature* 2015, 527: 329-335; *J Extracell. Vesicles* 2013, 2: 1-10; *Proc. Natl. Acad. Sci. U S A* 2016, 113: E968-977; *Proc. Natl. Acad. Sci. U S A* 2017, 114: 3175-3180; *Mol Cell Proteomics* 2013, 12: 343-355). Due to the natural origination, the exosomes exhibited low immunogenicity and long blood circulation time, therefore acting as an ideal drug carrier (*J Control Release* 2018, 289: 56-69; *Adv. Drug Delivery Rev.* 2013, 65: 391-397). For example, CD47 expressed in exosomes is the ligand for signal regulatory

protein alpha (SIRP α), and CD47-SIRP α binding initiates the 'don't eat me' signal that inhibits phagocytosis, resulting in the long blood circulation and enhanced tumor accumulation (*Nature* 2017, 546: 498; *Nanomedicine* 2018, 14: 195-204; *Biomaterials* 2017, 121:121–129).

In our manuscript, we found that CD54 expressed in tumor exosomes facilitates tumor accumulation (Supplementary Fig. 22 in the revised manuscript), tumor penetration (Supplementary Fig. 26 in the revised manuscript) and cellular uptake by cancer cells (Supplementary Fig. 19 in the revised manuscript). In addition, DOX@E-PSiNPs did not induce immune response in C57BL/6 mice, as evidenced by the fact that no significant change in the content of IgM, TNF- α , IL-1 β and IL-6 was detected in serum of DOX@E-PSiNPs-treated mice (Supplementary Fig. 37 in the revised manuscript).

4. The authors state the exosome-sheathed nano-delivery platform has improved tissue distribution and penetration due to EPR effect. The particle characterization, including size and surface charge, does not explain this apparent improved distribution. Have the authors considered mechanistic studies to determine critical factors that may enhance delivery in valid, realistic tumor microenvironments. Distribution mathematical modeling may be very helpful.

Response: We thank the reviewer's critical comments. In our manuscript, the size of E-PSiNPs and PSiNPs was 260 \pm 15 nm and 150 \pm 11 nm, and their zeta-potential was -11.0 \pm 0.4 mV and -10.8 \pm 0.2 mV, respectively. The small difference in size and zeta-potential might not significantly affect the tumor accumulation and penetration. We hypothesized that compared with PSiNPs, the most remarkable feature of E-PSiNPs was the exosomes coated on PSiNPs. The proteins on exosomes might affect the *in vivo* transport process of DOX@E-PSiNPs. For example, in the original manuscript, we showed that CD54, a member of the immunoglobulin supergene family, was found to be involved in the cross-reactive cellular uptake of DOX@E-PSiNPs by cancer cells, as evidenced by the fact that DOX@E-PSiNPs exocytosed from B16-F10 and H22 cells expressed CD54 (Supplementary Fig. 16A), and pretreatment with CD54 antibody decreased the cellular uptake of DOX@E-PSiNPs exocytosed from H22 cells by H22 and B16-F10 cells (Supplementary Fig. 16B).

To determine whether CD54 affect the tumor accumulation and penetration of DOX@E-PSiNPs, H22 tumor-bearing mice were intravenously injected with DOX@E-PSiNPs or DOX@E-PSiNPs pretreated with CD54 antibody. At 24 h after injection, the mice were sacrificed, and tumors and major normal organs (heart, liver, spleen, lung and kidney) were collected, followed by DOX content measurement. The results showed that pretreatment with CD54 antibody significantly decreased the tumor accumulation of DOX@E-PSiNPs (Supplementary Fig. 22 in the revised manuscript). Furthermore, confocal microscopic images showed that DOX@E-PSiNPs were distributed widely in whole section, while pretreatment with CD54 antibody resulted in a stronger co-localization of DOX@E-PSiNPs with FITC-CD31-labeled endothelial cells, suggesting that CD54 antibody decreased the tumor penetration of DOX@E-PSiNPs (Supplementary Fig. 26 in the revised manuscript). Therefore, our results show that CD54 expressed in tumor exosomes played an important role in mediating the tumor accumulation, tumor penetration and cross-reactive cellular uptake by cancer cells.

According to the reviewer's suggestion, we added these data in the revised manuscript, Supplementary Fig. 22 and 26.

Response to Reviewer #4:

Reviewer #4's feedback on rebuttal to ref#3's concerns from last round:

1. I have read the author's response to the concerns from reviewer #3, and I think the authors addressed most of the concerns, but it should be addressed why DOX@E-PSiNPs decreases P-gp expression in CSCs compared with PBS, free DOX and DOX@PSiNPs.

Response: We thank the editor and reviewer's critical question. P-gp is a plasma membrane transporter, whose expression was associated with cell membrane microenvironment (*Adv. Drug. Deliv. Rev.* 2013, 65: 1686–1698; *Proc. Natl. Acad. Sci. U S A.* 2002, 99: 10347–10352; *Cancer Res.* 2003, 63: 3084–3091). In the revised manuscript, atomic force microscopic (AFM) analysis showed that compared with H22 cancer stem cells (CSCs) treated with free DOX or DOX@PSiNPs, more obvious change in cell membrane morphology and roughness was detected in DOX@E-PSiNPs-treated H22 CSCs (Supplementary Fig. 11A,B in the revised manuscript), suggesting that DOX@E-PSiNPs have a stronger interaction with cell membrane. Furthermore, the stronger interaction between DOX@E-PSiNPs and cell membrane induced a significant decrease in cell membrane fluidity (Supplementary Fig. 11C in the revised manuscript) as measured by fluorescence polarization of 1,6-diphenyl-1,3,5-hexatriene (DPH) (*Langmuir* 2018, 34: 5097-5105; *Langmuir* 2013, 29: 4830-4838; *Biomaterials* 2011, 32: 5148-5157], which might induced a decrease in P-gp expression in H22 CSCs after treatment with DOX@E-PSiNPs.

According to the reviewer's suggestion, we added these results in the revised manuscript, Supplementary Fig. 11.

2. In addition, the question below was not well answered with the data support: "Can the authors offer any data explaining the lack of any complete responders? Is this due to CSCs developing a resistant phenotype?"

Response: We thank the reviewer's critical comments. We agreed with the reviewer that the lack of any complete responders might be due to the emergence of resistance in CSCs. To confirm this hypothesis, H22 tumor-bearing mice were intravenously injected with DOX@E-PSiNPs at the DOX dosage of 0.5 mg/kg, or co-injected with DOX@E-PSiNPs at the same DOX concentration and all-trans-retinoic acid (ATRA), a powerful

differentiating agent of CSCs (*Biomaterials* 2015, 37: 405-414; *Nat. Commun.* 2018, 9: 3390-3407) every three days for five times. The results showed that compared with only DOX@E-PSiNPs treatment group, co-injection of DOX@E-PSiNPs and ATRA resulted in a significant tumor inhibition, and the tumor ablation was observed in 2 tumor-bearing mice (Supplementary Fig. 32A-G in the revised manuscript, the total mouse number was 6 for each group). Correspondingly, less side population cells in tumor tissues (Supplementary Fig. 32H in the revised manuscript), less colony numbers (Supplementary Fig. 32I in the revised manuscript) and smaller colony sizes (Supplementary Fig. 32J in the revised manuscript) after seeding the tumor cells in 3D fibrin gels were observed in co-injection of DOX@E-PSiNPs and ATRA group, suggesting that CSCs might be responsible for the drug resistance. Furthermore, we treated H22 tumor-bearing mice with DOX@E-PSiNPs at DOX dosage of 0.8 mg/kg. The results showed that increasing the DOX dosage also improved the treatment efficacy of DOX@E-PSiNPs (3 tumor ablation in 6 mice) and increasing the CSCs killing activity.

Taken together, we can combine DOX@E-PSiNPs with differentiating agent of CSCs, or increase the used dosage of DOX@E-PSiNPs to decrease the CSCs, generating stronger anticancer activity. Our results also showed that combination treatment of DOX@E-PSiNPs and ATRA, or increasing the used dosage of DOX@E-PSiNPs is safe, as evidenced by body weight, routine blood test and serological analysis (Supplementary Fig. 33 in the revised manuscript).

According to the editor and reviewer's suggestion, we added these data in the revised manuscript, Supplementary Fig. 32, 33.

REVIEWERS' COMMENTS:

Reviewer #1 (Remarks to the Author):

The authors have provided reasonable explanations to the prior comments, while not most directly. Some work maybe carried out in the future. Overall, I support publication of this manuscript in the current form.

Reviewer #3 (Remarks to the Author):

My comments have been appropriately and satisfactorily addressed by the authors

Response to Reviewer 1

The authors have provided reasonable explanations to the prior comments, while not most directly. Some work may be carried out in the future. Overall, I support publication of this manuscript in the current form

Response: We thank the reviewer's comments on our manuscript. We will carry out some experiments to further explore the mechanism on the exocytosis of exosome-coated PSiNPs in the future.

Response to Reviewer 3

My comments have been appropriately and satisfactorily addressed by the authors

Response: We thank the reviewer's comments on our manuscript.